# Improving Local Explainability By Learning Causal Graphs From Data

**Daan Roos**  *d.f.a.roos@uva.nl*
*UvA-Bosch Delta Lab,*
*University of Amsterdam*

**Sebastian Gerwinn**  *Sebastian.Gerwinn@de.bosch.com*
*Bosch Center for Artificial Intelligence,*
*Robert Bosch GmbH*

**Jan-Willem van de Meent**  *j.w.vandemeent@uva.nl*
*UvA-Bosch Delta Lab,*
*University of Amsterdam*

**Sara Magliacane**  *sara.magliacane@gmail.com*
*Causal Machine Learning group, Saarland University, &*
*Amsterdam Machine Learning Lab, University of Amsterdam*

**Reviewed on OpenReview:** *https://openreview.net/forum?id=A1bXT7RQLU*

## Abstract

Causal Shapley values take into account causal relations among dependent features to adjust the contributions of each feature to a prediction. A limitation of this approach is that it can only leverage known causal relations. In this work we combine the computation of causal Shapley values with causal discovery, i.e., learning causal graphs from data. In particular, we compute causal explanations across the Markov Equivalence Class (MEC), a set of candidate causal graphs learned from observational data, providing a list of causal Shapley values that explain the prediction. We propose two methods for estimating this list efficiently, drawing on the equivalences of the interventional distributions for a subset of the causal graphs. We evaluate our methods on synthetic and real-world data, showing that they provide explanations that are more consistent with the true causal effects compared to traditional Shapley value approaches that disregard causal relations. Our results show that even when the Markov Equivalence Class is learned incorrectly, in most settings the explanations of our framework are on average closer to true causal Shapley values than marginal and conditional Shapley values.

## 1 Introduction

The advancement of machine learning models has ushered in a new era of AI capabilities, surpassing simpler models in performance. However, this increase in complexity brings a significant challenge: a marked decrease in interpretability with advanced models often resembling a black box (Zhang et al., 2020; Carvalho et al., 2019). This opacity is not just a technical concern, but it becomes a matter of ethical and legal importance in high-stakes applications, e.g., medical diagnostics or autonomous vehicles, where the inability to explain a model's reasoning can lead to issues of trust and accountability (Doshi-Velez & Kim, 2017).

Shapley values are a notable method used to explain how machine learning models make their decisions (Zhou et al., 2021). They offer a principled local approach to attributing the output of a model to its input features, breaking down the prediction into contributions from each feature. When using Shapley values for explainability, various methods assume independence between the model's input features (Štrumbelj &

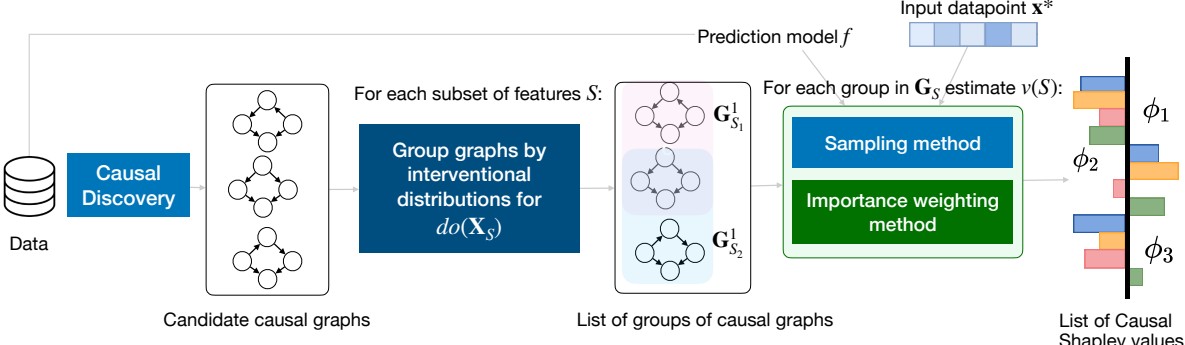

Figure 1: Overview of our MEC Shapley framework that provides a range of causal Shapley values, describing the causal impact of a feature on an input datapoint, even when the causal graph is unknown. We first apply causal discovery to learn a set of candidate causal graphs directly from the data. For each subset of features $S$ that we consider, we group these causal graphs based on sharing the same interventional distribution for an intervention on $\mathbf{X}_S$ in a list of groups $\mathbf{G}_S$. For each group of graphs, we can then estimate the value function $v(S)$ for the input datapoint $\mathbf{x}^*$ given the prediction model $f$ with one of our two methods, a sampling methods or an importance weighting method, which allows us to provide a list of causal Shapley values that represent a range of explanations for the input datapoint that reflect the different candidate causal relations between the features.

Kononenko, 2014; Lundberg & Lee, 2017). As a result, importance is only attributed to features that directly contribute to a prediction (Heskes et al., 2020). As a consequence, many inputs to the model will be off the data manifold (Frye et al., 2021) and may lead to misleading explanations (Aas et al., 2021). Relaxing this assumption leads to explanations where correlated features share importance, regardless of their causal effect on the prediction (Aas et al., 2021). As a result, conditional Shapley values may attribute importance to features that are merely correlated with the prediction.

Heskes et al. (2020) integrate causal modeling with the Shapley value framework, leveraging the knowledge of causal relations among the features to adjust the contribution of each feature. This ensures that features only contribute to explanations when they have a causal effect on the prediction, whether direct or indirect. This paper aims to address one of the key challenges in applying causal Shapley values: leveraging only known causal relations. As described in the overview in Fig. 1, we first use causal discovery methods, e.g. PC (Spirtes et al., 2000) and FGES (Chickering, 2003; Ramsey et al., 2017), to learn an equivalence class of causal graphs directly from data and then combine it with the causal Shapley value estimation in an efficient way. This process provides a set of explanations that reflect different possible causal structures, thereby representing causal uncertainty. Our contributions are as follows:

- We introduce a framework for combining causal discovery with causal Shapley values (Heskes et al., 2020), extending their applicability beyond known causal relations. In particular, we address the challenge that causal discovery methods return a set of candidate causal graphs and hence our output is a list of causal Shapley values.
- We propose two methods to efficiently estimate causal Shapley values using only observational data, a sampling method based on simulating interventional data and an importance weighting method. Both of these methods leverage the ID algorithm (Shpitser & Pearl, 2006) to identify the estimand formula for interventional distributions and hence reuse computation across the set of graphs returned by causal discovery methods. Additionally, the importance weighting method only requires the prediction model to be applied on the initial data, thus reducing the computational load.
- We evaluate the performance of our methods using simulated and real-world data. Interestingly, we find that even when the true causal graph or its equivalence class are not learned correctly, the explanations returned by our method tend to be closer on average to the true causal Shapley value than the marginal and conditional Shapley values.

## 2   Background

In local explainability, the goal is to explain the prediction of a model $f$ for a single input $\mathbf{x}^*$ (Doshi-Velez & Kim, 2017) that is an instantiation of the features $\mathbf{X} = \{X_1, \ldots, X_p\}$ with distribution $P(\mathbf{X})$. A popular method for explaining the influence of each feature $\mathbf{x}_i^*$ on the model output $f(\mathbf{x}^*)$ are Shapley values, (Shapley, 1953), introduced in game theory as a way of distributing a value over a set of $p$ 'players'. The Shapley value of player $i$ for a value function $v$ is:

$$\phi_i(v) = \frac{1}{p} \sum_{S \subseteq [p] \setminus \{i\}} \binom{p-1}{|S|}^{-1} v(S \cup \{i\}) - v(S), \quad i \in [p], \tag{1}$$

where $[p] = \{1, \ldots, p\}$ is the set of all players and each $S$ represents a subset of $[p] \setminus \{i\}$, called a *coalition*. This value can be seen as a weighted average of the marginal contribution of $i$ to every possible coalition of other players $S$. It can be shown that the contribution of each player sums to the value gained when every player contributes, which ensures that the value produced by the whole group together is distributed over the player's individual Shapley values.

To apply Shapley values in local explainability, we consider each feature a "player". It is not obvious how to choose an appropriate value function $v$ and many options have been proposed (Sundararajan & Najmi, 2020). In particular, we need to choose what happens to features that are in the coalition, denoted by the index set $S$, and features that are not in the coalition, denoted by the index set $\overline{S}$. A common choice for $v$ is the expectation of our model w.r.t. features that are not in the coalition. If we assume independent features, the value function is the expectation w.r.t. the marginal distribution of the features out of the coalition (Štrumbelj & Kononenko, 2014; Datta et al., 2016; Lundberg & Lee, 2017). The value function is then:

$$v(S) = \mathbb{E}_{P(\mathbf{X}_{\overline{S}})}[f(\mathbf{x}_S^*, \mathbf{X}_{\overline{S}})], \tag{2}$$

where $\mathbf{x}_S^*$ are the features of the input that are in $S$, while $\mathbf{X}_{\overline{S}}$ represents the features that are not in $S$ over which we take the expectation. We refer to the Shapley values based on this value function as *marginal* Shapley values. Aas et al. (2021) take into account dependence between the features with the following value function, which we refer to as *conditional* Shapley values,

$$v(S) = \mathbb{E}_{P(\mathbf{X}_{\overline{S}} | \mathbf{X}_S = \mathbf{x}_S^*)}[f(\mathbf{x}_S^*, \mathbf{X}_{\overline{S}})]. \tag{3}$$

Heskes et al. (2020) propose *causal* Shapley values, which take into account causal relations between features, represented as a Structural Causal Model (Pearl, 2009; Spirtes et al., 2000), by replacing conditional distributions with *interventional distributions*. This results in choosing $v$ as follows:

$$v(S) = \mathbb{E}_{P(\mathbf{X}_{\overline{S}} | do(\mathbf{X}_S = \mathbf{x}_S^*))}[f(\mathbf{x}_S^*, \mathbf{X}_{\overline{S}})]. \tag{4}$$

Here $P(\mathbf{X}_{\overline{S}} | do(\mathbf{X}_S = \mathbf{x}_S^*))$, represents the distribution after performing an intervention $do(\mathbf{X}_S = \mathbf{x}_S^*)$ that forces $\mathbf{X}_S$ to take the value $\mathbf{x}_S^*$. This distribution is in general not the same as the conditional distribution $P(\mathbf{X}_{\overline{S}} | \mathbf{X}_S = \mathbf{x}_S^*)$ (Pearl, 2009). Heskes et al. (2020) show that with this value function, an indirect effect on the contribution of a feature only occurs when a change in a feature's value has a causal effect on the prediction, which leads to an explanation that more accurately reflects the effect of each feature. We provide a short introduction to causality in App. A.1 and an comparison between conditional and causal Shapley values, including a worked out example in our causal discovery setting in App. A.2.1.

## 3   MEC Shapley Values

In this paper, we extend the causal Shapley value framework from leveraging known causal relations to the case in which we learn a set of possible causal graphs from observational data, i.e. a Markov Equivalence Class (MEC), using causal discovery methods. This results in a set of causal explanations, one for each Markov-equivalent causal graph that may have generated the data. Considering a set of causal models compounds the computational challenges associated with computing Shapley values. After estimating the

MEC, a naive strategy is to iterate over each Directed Acyclic Graph (DAG) in the MEC and compute the corresponding causal Shapley value. However, the size of a MEC grows rapidly with the amount of nodes (He et al., 2015). We therefore want to reuse computation where possible by identifying which DAGs share the same interventional distribution for a subset of features $S$ and by introducing two methods to estimate the interventional expectation: (i) a sampling method and (ii) an importance weighting method. We describe each step of our pipeline in the following.

## 3.1 Causal discovery and grouping of graphs

Given a dataset $\mathcal{D} = \{\mathbf{x}^{(k)}\}_{k=1}^{D}$ for features $\{\mathbf{X}\} = \{X_1, \ldots, X_p\}$, our goal is to explain the prediction of a prediction model $f$ for input data point $\mathbf{x}^*$. The first step is learning causal relations between the features from the data $\mathcal{D}$ with causal discovery. We assume that the *causal Markov* and *faithfulness assumptions* hold, i.e. that the conditional independences in the observational data correspond to d-separations in the true causal graph (Spirtes et al., 2000). We assume that we are in a *causally sufficient* setting, in which we do not have unobserved confounding or selection bias and the causal graph is a Directed Acyclic Graph (DAG). The output of most causal discovery methods is a Markov Equivalence Class (MEC), i.e. all graphs that induce the same conditional independences and dependences as in the observational data. We denote the MEC as $\mathbf{G}$ and define it as a set of candidate graphs $\mathbf{G} = \{G_1, \ldots, G_c\}$. For many sets $S$, multiple graphs in the MEC share the interventional distribution $P(\mathbf{X}_{\overline{S}}|do(\mathbf{X}_S = \mathbf{x}_S^*))$. By identifying which graphs share this distribution we only need to sample from one of them to estimate $v(S) = \mathbb{E}[f(\mathbf{X}_{\overline{S}}, \mathbf{x}_S^*)|do(\mathbf{X}_S = \mathbf{x}_S^*)]$. So for each intervention target $S$ we rewrite the interventional distributions with the ID algorithm (Shpitser & Pearl, 2006) as an estimand of observational quantities. We group graphs with the same estimand for the intervention on $S$ in a list of set of graphs $\mathbf{G}_S$, where for each group we estimate $v(S)$ once.

## 3.2 Sampling method

After learning the MEC and grouping the graphs based on their interventional estimands, as described in Sec. 3.1, we can now use a sampling method for estimating causal Shapley values, similar to the method used by Heskes et al. (2020) for a single, known causal graph. For each subset of features $S$ and a graph $G_i$ in each group of $\mathbf{G}_S$, we follow three steps: (i) conditional distribution estimation, (ii) simulation of interventional data, and (iii) estimation of the value function.

First, we consider $G_i$ as a potential causal graph and follow the topological order of $G_i$ to estimate $P_{G_i}(X_j|\mathbf{X}_{\mathrm{Pa}_{G_i}(j)})$ for each variable $X_j$, where $\mathbf{X}_{\mathrm{Pa}_{G_i}(j)}$ are the variables in $\mathbf{X}$ that are the causal parents of $\mathbf{X}_j$ in $G_i$. We then generate simulated interventional samples from the interventional distribution $P_{G_i}(\mathbf{X}|do(\mathbf{X}_S = \mathbf{x}_S^*))$ that assumes that $G_i$ is the true causal graph. In particular, we set all of the values of $\mathbf{X}_S = \mathbf{x}_S^*$ and use the truncated factorization formula (Eq. 11 in App. A.1) for which we sample each factor in a topological order in $G_i$ as follows:

$$\{\hat{\mathbf{x}}^{(k,G_i,S)}\}_{k=1}^{n_{mc}} \sim \prod_{j \notin S} P_{G_i}(X_j|\mathbf{X}_{\mathrm{Pa}_{G_i}(j)})\delta(\mathbf{X}_S = \mathbf{x}_S^*)$$
$$= P_{G_i}(\mathbf{X}|do(\mathbf{X}_S = \mathbf{x}_S^*)),$$

where $\delta$ is the Dirac delta function, forcing $\mathbf{X}_S$ to take the values $\mathbf{x}_S^*$. We then use the $n_{mc}$ simulated samples in a Monte-Carlo approximation of the value function $v$ for the graph $G_i$ as follows:

$$v(S)^{G_i} = \mathbb{E}[f(\mathbf{x}_S^*, \mathbf{X}_{\overline{S}})|G_i] \approx \frac{1}{n_{\mathrm{mc}}} \sum_{k=1}^{n_{\mathrm{mc}}} f(\hat{\mathbf{x}}^{(k,G_i,S)}).$$

Having estimated each interventional expectation, we combine them using Eq. 1 or, when sampling only a subset of all possible subsets, using Kernel SHAP (Lundberg & Lee, 2017; Aas et al., 2021).

## 3.3 Importance Weighting (IW) method

The sampling method from Sec. 3.2 may incur high computational cost, since it simulates new datapoints for each graph and intervention, each of which must be evaluated by a potentially expensive model. To

improve the computational efficiency, we introduce an importance-weighting method that re-uses a single set of predicted samples for an intervention on $\mathbf{x}_S$ across all DAGs in the MEC. The pseudocode for the method is provided in Alg. 1. While this method avoids having to evaluate new simulated data points, this approach can be susceptible to high variance if the target interventional distribution differs significantly from the observational data, a known limitation of importance sampling that the sampling-based method avoids.

We start by defining the set of subsets of feature that we consider $\mathbf{S}$ similar to KernelSHAP (Lundberg & Lee, 2017) (Lines 1-2). We learn the MEC on the initial data $\mathcal{D}$ using standard causal discovery methods and group the graphs based on their interventional estimands, as described in Sec. 3.1 (Lines 3-9). For each subset of features $S \in \mathbf{S}$ we then compute the value function $v(S)$ for each group of graphs (Lines 10-19), which we describe in detail in the following, and apply KernelSHAP (Lundberg & Lee, 2017) following Aas et al. (2021) to compute the final Shapley values (Lines 20-24).

The core of this algorithm is the computation of the value functions for each group of graphs in an efficient way. The first step is to compute the predictions $f(\mathbf{x}_{\overline{S}}^{(k)}, \mathbf{x}_S^*)$ using the prediction model $f$ and a combination of the initial data $\mathcal{D}$ and the input datapoint $\mathbf{x}^*$ (Line 11). We reweight these predictions based on the interventional distribution in each graph $G_i$ and estimate the expected value $\mathbb{E}_{P_{G_i}(\mathbf{X}_{\overline{S}}|do(\mathbf{X}_S=\mathbf{x}_S^*))}[f(\mathbf{x}_S^*, \mathbf{X}_{\overline{S}})|G_i]$, which is the value function used in causal Shapley values (Heskes et al., 2020). We rewrite $v(S)$ for a graph $G_i$ as a weighted expectation over samples from the marginal distribution $P(\mathbf{X}_{\overline{S}})$, where $\overline{S}$ are the features not in $S$:

$$v^{G_i}(S) = \mathbb{E}_{P_{G_i}(\mathbf{X}_{\overline{S}}|do(\mathbf{X}_S=\mathbf{x}_S^*))}[f(\mathbf{x}_S^*, \mathbf{X}_{\overline{S}})|G_i] \tag{5}$$

$$= \int f(\mathbf{x}_S^*, \mathbf{X}_{\overline{S}})P_{G_i}(\mathbf{X}_{\overline{S}}|do(\mathbf{X}_S = \mathbf{x}_S^*))\frac{P(\mathbf{X}_{\overline{S}})}{P(\mathbf{X}_{\overline{S}})}d\mathbf{X}_{\overline{S}} \tag{6}$$

$$= \mathbb{E}_{P(\mathbf{X}_{\overline{S}})}\left[f(\mathbf{x}_S^*, \mathbf{X}_{\overline{S}})\frac{P_{G_i}(\mathbf{X}_{\overline{S}}|do(\mathbf{X}_S = \mathbf{x}_S^*))}{P(\mathbf{X}_{\overline{S}})}\right], \tag{7}$$

where in Eq. 6 we introduce $P(\mathbf{X}_{\overline{S}})$ in the denominator and the numerator. To approximate this expectation, we first estimate the conditional densities $\hat{P}_{G_i}(X_j|\mathbf{X}_{\text{Pa}_{G_i}(j)})$ from data $\mathcal{D}$ (Line 14). We then re-weight each prediction $f(\mathbf{x}_{\overline{S}}^{(k)}, \mathbf{x}_S^*)$ for each factorization in a Monte Carlo approximation of Eq. 7 as follows:

$$v^{G_i}(S) \approx \frac{1}{D}\sum_{k=1}^{D} f(\mathbf{x}_{\overline{S}}^{(k)}, \mathbf{x}_S^*)\frac{\hat{P}_{G_i}(\mathbf{X}_{\overline{S}}|do(\mathbf{X}_S = \mathbf{x}_S^*))}{\hat{P}(\mathbf{X}_{\overline{S}})}, \tag{8}$$

$$\hat{P}_{G_i}\left(\mathbf{X}_{\overline{S}}|do(\mathbf{X}_S = \mathbf{x}_S^*)\right) = \prod_{j\in\overline{S}} \hat{P}_{G_i}(X_j|\mathbf{X}_{\text{Pa}_{G_i}(j)\cap\overline{S}}, \mathbf{x}_{\text{Pa}_{G_i}(j)\cap S}^*)$$

following the truncated factorization formula (Eq. 11), where we force all variables in $S$ to take the value $\mathbf{x}_S^*$, including the parents of each variable $X_j$ in $G_i$, represented as $\text{Pa}_{G_i}(j) \cap S$. We compute Eq. 8 for a single graph in each group and assign it to the other graphs in the same group (Lines 15-16). For each graph $G_i$ we use $v(S)^{G_i}$ for each set of features $S$ to compute the Kernel SHAP values (Lundberg & Lee, 2017) (Lines 19-22). We return a list of Kernel SHAP values, one for each DAG in the MEC $\mathbf{G}$ (Line 23).

## 4 Related work

Connections between causality and explainability have been explored in various works (Beckers, 2022; Karimi et al., 2023). Many of these focus on leveraging causal knowledge to improve explainability (Khademi & Honavar, 2020; Schwab & Karlen, 2019; Oesterle et al., 2023), but they do not use the framework of Shapley values, which provides desirable properties and computational tools in terms of interpretability and efficiency. Some approaches employ causal discovery for explainability outside the Shapley framework. For example, Sani et al. (2020) use causal discovery for explainability of black-box methods. In contrast to our work, they consider unstructured data such as images and allow for potential latent confounding, but do not quantify causal effects. Similarly, Takahashi et al. (2024) also perform causal discovery for local explanations, but focus on the context of counterfactual explanations.

---

**Algorithm 1** Importance weighting method for Estimating MEC Shapley Values

---

**Input:** Dataset $\mathcal{D} = \{\mathbf{x}^{(k)}\}_{k=1}^{D}$, prediction model $f$, target variable $Y$, data point to explain $\mathbf{x}^*$, number of combinations to consider $n_{\text{comb}} \leq |2^{[p]}|$ for $p$ features.

1: Initialize set of subsets of features to consider $\mathbf{S} \leftarrow \{\emptyset, [p]\}$ following (Lundberg & Lee, 2017).
2: Sample $n_{\text{comb}} - 2$ subsets of the power set $2^{[p]} \setminus \{\emptyset, [p]\}$ with probability equal to kernel weights $k(S) = (p-1)/\left(\binom{p}{|S|}|S|(p-|S|)\right)$ and add them to $\mathbf{S}$.
3: Learn the MEC $\mathbf{G} = \{G_1, \ldots, G_c\}$ by applying a causal discovery method to $\mathcal{D}$
4: **for** each subset $S \in \mathbf{S}$ of features **do**
5:     **for** each graph $G_j \in \mathbf{G}$ **do**
6:         Identify estimand formula for $P^{G_j}(\mathbf{X}_{\overline{S}}|do(\mathbf{X}_S = \mathbf{x}_S^*))$ with ID algorithm.
7:     **end for**
8:     Group graphs with same estimand formula for the intervention on $S$ in a set of graphs $\mathbf{G}_S^l$.
9: **end for**
10: **for** each subset $S \in \mathbf{S}$ of features **do**
11:     Compute predictions $\{f(\mathbf{x}_{\overline{S}}^{(k)}, \mathbf{x}_S^*)\}_{k=1}^{D}$ with observational data $\{\mathbf{x}_{\overline{S}}^{(k)}\}_{k=1}^{D}$ and input $\mathbf{x}_S^*$.
12:     **for** each group of graphs $\mathbf{G}_S^l$ with the same estimand formula for $P(\mathbf{X}_{\overline{S}}|do(\mathbf{X}_S = \mathbf{x}_S^*))$ **do**
13:         Choose any $G_j$ in $\mathbf{G}_S^l$.
14:         Estimate conditional probabilities $\hat{P}_{G_i}(X_j|\mathbf{X}_{\text{Pa}_{G_i}(j)})$ from observational data $\mathcal{D}$.
15:         Calculate value of $v$ for graph $G_j$ and set $S$ with Eq. 8:

$$v^{G_j}(S) \leftarrow \frac{1}{D} \sum_{k=1}^{D} \frac{\prod_{h \notin S} \hat{P}(\mathbf{x}_h^{(k)}|\mathbf{x}_{\text{Pa}_{G_j}(h) \cap \overline{S}}^{(k)}, \mathbf{x}_{\text{Pa}_{G_j}(h) \cap S}^*)}{\hat{P}(\mathbf{x}_{\overline{S}}^{(k)})} f(\mathbf{x}_{\overline{S}}^{(k)}, \mathbf{x}_S^*)$$

16:         Store the value of $v^{G_j}(S)$ for the value functions of all other graphs in $\mathbf{G}_S^l$.
17:     **end for**
18: **end for**
19: **for** each graph $G_j \in \mathbf{G}$ **do**
20:     Collect all value functions for each subset $S \subseteq \mathbf{S}$ in a vector $\mathbf{v}^{G_j} \leftarrow (\mathbf{v}^{G_j}(\emptyset), \ldots, \mathbf{v}^{G_j}([p]))$.
21:     Compute the Kernel SHAP value $\phi^{G_j}$ for $\mathbf{S}$ with $\mathbf{v}^{G_j}$ following (Lundberg & Lee, 2017).
22: **end for**
23: **return** List of Kernel SHAP values $\{\phi^{G_j}\}_{G_j \in \mathbf{G}}$ for each DAG in the MEC $\mathbf{G}$.

---

Several works have combined Shapley values with causal reasoning. This work is closely related to that of Jung et al. (2022), who adopt the same value function in the context of causal contribution analysis, and propose a weighted estimator to estimate the value functions in the case of discrete data. Janzing et al. (2020) argue that, since we are explaining an algorithm, the features should be seen as independent, such that interventional Shapley values become equivalent to marginal Shapley values. As shown by Heskes et al. (2020), this leads to explanations that do not consider indirect effects of the features on the predictions, e.g., when we use a proxy of a feature instead of the feature itself, as discussed also by Frye et al. (2020) in the context of *unresolved discrimination.*

Frye et al. (2020) introduce asymmetric Shapley values, incorporating causal knowledge into explanations by weighting subsets according to their agreement with a partial causal ordering. They do not make use of the concepts of interventions, but as noted by Heskes et al. (2020) they can be combined with causal Shapley values, specifically when making use of chain graphs. Budhathoki et al. (2022) and Strobl & Lasko (2022) apply causality to Shapley values, but in the context of root cause analysis, specifically to explain outliers instead of general predictions by any arbitrary model given a known causal graph. These works only leverage known causal relations, either as a partial causal ordering or a complete causal graph. An exception is the work by Strobl & Lasko (2022) that learn the causal graph, but assume a linear non-Gaussian model, a more specific setting than ours which leads to a single causal graph instead of a MEC. Concurrently to our work, Ng et al. (2025) propose Causal SHAP, which integrates the PC and IDA algorithms into SHAP to yield a

single attribution vector corrected by estimated causal strengths. In contrast, our framework is agnostic to the specific causal discovery method and explicitly considers the entire Markov equivalence class, producing a range of causal Shapley values that reflect multiple plausible causal structures.

## 5 Experiments

We evaluate our methods across synthetic datasets and two real-world datasets. One of the challenges when evaluating methods for explainability is that we lack agreed-upon metrics for assessing the quality of explanations. We therefore focus on a tangible, well-defined objective: evaluating how well and how efficiently our methods approximate the theoretical causal Shapley values derived from a ground-truth causal graph. These values provide a principled framework for attributing a model's prediction to its input features based on their causal effects.

In the synthetic datasets we have access to the true interventional distributions, allowing us to compare the output of our methods to the true causal Shapley values. We evaluate in a linear Gaussian and a non-linear setting. In addition to comparing with the ground truth causal Shapley value defined by the true interventional distributions, we also compare the outcome of our method with the marginal and conditional Shapley values, which do not take any causal relations into account. To implement our method we adapt the `shapr` package by Aas et al. (2021). For causal discovery, we used the `pcalg` package (Kalisch et al., 2012) and the `Tetrad` toolbox (Ramsey et al., 2018). For identifying the interventional estimand formulas we use the `causaleffect` package (Tikka & Karvanen, 2017a;b). For prediction we use XGBoost models (Chen & Guestrin, 2016). We provide the code to reproduce our experiments in the supplementary material.

### 5.1 Synthetic data

We generate data for two settings, linear Gaussian and non-linear causal models. For each setting, we consider $n_{\mathrm{scm}} = 40$ randomly generated causal models for which we generate synthetic data as follows.

For each model we first sample an Erdos-Renyi graph of size $p = 5, 10, 15$ where we specify the expected number of neighbors $d = 1, 2, 3$ for each node. We add our 'dependent' node $Y$, the target variable for our prediction model, for which we specify the minimum number of incoming edges ($m_{\min} = 2, 3, 5$) and the expected number of incoming edges ($\bar{m} = 3, 6, 9$).

In the linear Gaussian setting, nodes are parametrized as $X_i = \sum_{j \in \mathrm{Pa}_i} w_{ij} X_j + \epsilon_i$ where $w_{ij} \sim U((-2, -0.5) \cup (0.5, 2))$ and $\epsilon_i = \mathcal{N}(0, 1)$. For the non-linear setting we simulate data with the equations $X_i = w_2^i \sigma(\sum_{j \in \mathrm{Pa}_i} w_1^{ij} X_j) + \epsilon_i$ where $\sigma$ denotes the sigmoid function, $w_1^{ij} \sim U((-1.5, -0.5) \cup (0.5, 1.5))$, $w_2^i \sim U((-3, 1) \cup (1, 3))$, and $\epsilon_i$ follows either a $\mathcal{N}(0, 1)$ or a $U(-1, 1)$ distribution with equal probability. We apply causal discovery to standardized data. For each causal model we simulate $n_{\mathrm{test}} = 40$ data points $\mathbf{x}^*$ for which we compute the Shapley values.

**Methods.** For each model, following Heskes et al. (2020) we train an XGBoost model (Chen & Guestrin, 2016) on a subset of $n_{\mathrm{train}} = 10K$ samples for 100 rounds $n_{\mathrm{test}} = 40$ data points $\mathbf{x}^*$ which are unseen by the model. We compare the results of our methods with Marginal (Lundberg & Lee, 2017) and Conditional Shapley values (Aas et al., 2021). For the Conditional Shapley baseline, we estimate the value function $v(S) = \mathbb{E}[f(x) \mid X_S = x_S^*]$ by modeling the feature distribution $P(X)$ as a multivariate Gaussian, where the conditional distributions $P(X_{\bar{S}} \mid X_S)$ are derived analytically from the observational sample mean and covariance. This approach, implemented via the `shapr` package (Aas et al., 2021), serves as the standard baseline for dependent features but assumes linear relations between variables. We use the same conditional estimation for conditional and MEC Shapley values for a fair comparison. To evaluate whether our method provides benefits beyond improving conditional estimation, we additionally compare against conditional Shapley values estimated with Gaussian Copula and Conditional Inference Trees in App. A.3.3.

As causal discovery methods, we use the PC algorithm (Spirtes et al., 2000) and the FGES algorithm (Ramsey et al., 2017). We call the combination of our method with PC the *PC MEC* Shapley method, while the combination with FGES is the *FGES MEC* Shapley method. We compare the results of these algorithms with a causal discovery oracle, which we call the *Oracle MEC* method. Crucially, this is a partial oracle: it is

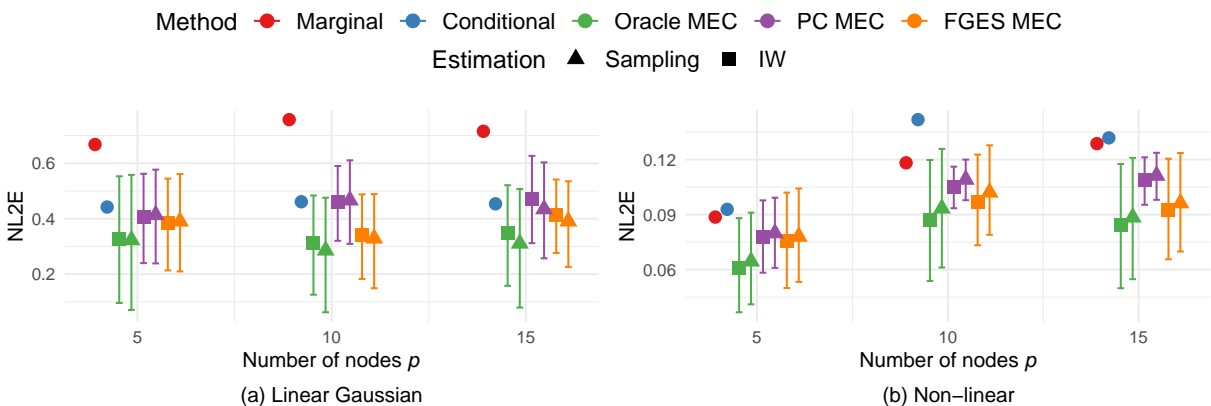

Figure 2: Normalized L2 error between the ground truth causal Shapley value and the Shapley values returned by each method for different parameters: (a) linear Gaussian setting with varying number of features $p$, (b) nonlinear Gaussian setting with varying number of features $p$, (c) linear Gaussian setting with varying number of expected neighbors $d$, and (d) linear Gaussian setting with varying expected number of nodes influencing $Y$ $\bar{m}$. For all MEC Shapley methods we plot the NL2E over the range of explanations, i.e., the minimum and maximum of NL2E across all DAGs $G_i$ in the MEC $\mathbf{G}$, averaged over all $n_{scm} = 40$ causal models. The marker represents the average NL2E across the DAGs in the MEC and models.

given the true underlying graph structure (and thus the true MEC), but it must still estimate the required conditional distributions from the same $n_{\text{obs}}$ samples as the other methods. For all methods, we consider a Sampling version, described in Sec. 3.2, and an IW version, described in Sec. 3.3.

We apply the PC algorithm to the linear data using partial correlation tests and for non-linear data we apply kernel-based conditional independence tests (Zhang et al., 2011), both with significance threshold $\alpha = 0.05$. We apply FGES in both settings using the BIC criterion. For the conditional and MEC Shapley methods, we model conditional distributions using a Gaussian parametric assumption. In the non-linear setting, we additionally evaluate conditional Shapley values using two more flexible conditional estimators (Gaussian copula and conditional inference trees) in App. A.3.3. We approximate marginal Shapley values using samples from empirical marginal distribution $p^*(\mathbf{X}_{\overline{S}})$. By default we use $n_{\text{cd}} = 1000$ observational samples for PC and FGES, $n_{\text{obs}} = 1000$ samples to estimate conditional expectations and $n_{\text{mc}} = 1000$ samples to estimate the Monte Carlo approximation. For $p = 5, 10$ we consider the number of subsets $n_{\text{comb}} = 2^p$, meaning that we use the full set of of combinations $S$. As the set of combinations increases exponentially with the amount of nodes, for $p = 15$ we set $n_{\text{comb}} = 8192$ for the ground truth and $n_{\text{comb}} = 4096$ for the other methods. We provide ablations for these parameters in App. A.3.

**Metrics.** We evaluate how close are explanations from different methods in terms of agreement with the ground truth causal Shapley values $\phi_{\text{true}}^{G^*}$, which we compute using the ground truth graph $G^*$ and sampling from the true interventional distributions with $n_{\text{mc}}^* = 4000$ Monte Carlo samples for each datapoint $\mathbf{x}^*$.

We then compare on the L2 error between the Shapley value returned by each method and the ground truth causal Shapley value, normalized based on the its value for the datapoint to allow comparison across different data points, defining the normalized L2 error as follows:

$$NL2E_{\text{method}} = \frac{\|\phi_{\text{true}}^{G^*} - \phi_{\text{method}}\|_2}{\|\phi_{\text{true}}^{G^*}\|_2} \tag{9}$$

where $\phi_{\text{true}}^{G^*}$ refers to the ground truth causal Shapley value for $\mathbf{x}^*$ and $\phi_{\text{method}}$ is the Shapley value returned by each method. Since MEC Shapley values methods provide a list of possible explanations, one for each causal graph in the MEC $\mathbf{G} = \{G_1, \ldots, G_c\}$, this results in a set of explanations $\{NL2E_{G_1}, \ldots, NL2E_{G_c}\}$. Note that this is also obviously true for the Oracle MEC Shapley values method, since the ground truth MEC includes multiple DAGs. To visualize this compactly our results, we consider the minimum and maximum NL2E across

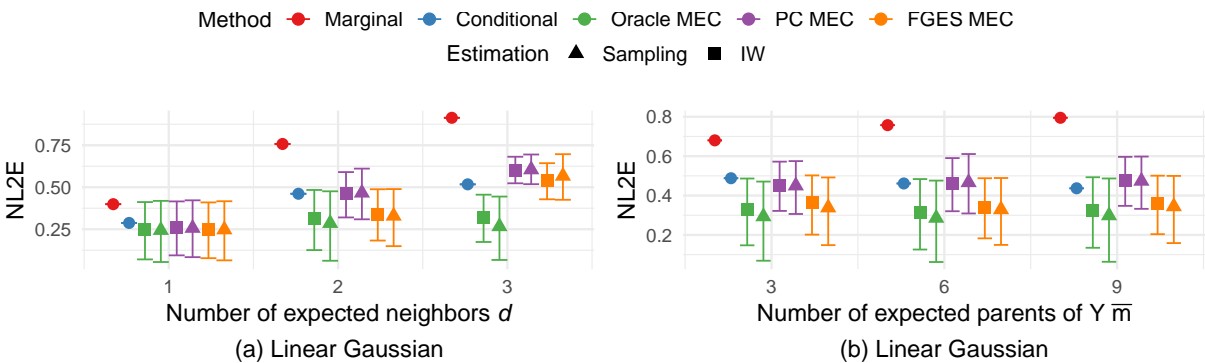

Figure 3: Normalized L2 error between the ground truth causal Shapley value and the Shapley values returned by each method for different parameters: (a) linear Gaussian setting with varying number of expected neighbors $d$, and (b) linear Gaussian setting with varying expected number of nodes influencing $Y$ $\bar{m}$. For all MEC Shapley methods we plot the NL2E over the range of explanations, i.e., the minimum and maximum of NL2E across all DAGs $G_i$ in the MEC $\mathbf{G}$, averaged over all $n_{scm} = 40$ causal models. The marker represents the average NL2E across the DAGs in the MEC and models.

all DAGs in a MEC as the *NL2E over the range of the explanations*, i.e., $[\min_{G_i \in \mathbf{G}} \mathrm{NL2E}_{G_i}, \max_{G_i \in \mathbf{G}} \mathrm{NL2E}_{G_i}]$. Moreover, we report the *average NL2E over the DAGs in the MEC*, i.e., $E_{G_i \in \mathbf{G}} \mathrm{NL2E}_{G_i}$.

**Results.** We report the results for all the synthetic data settings in the following. Fig. 2a shows the NL2E between each method and the ground truth causal Shapley as a function of the number of nodes $p = 5, 10, 15$ for linear Gaussian causal models for $d = 2$ and $\bar{m} = 3, 6, 9$.

For MEC Shapley values methods, the bars represent the NL2E over the range of explanations, so the minimum and maximum NL2E for each DAG in a MEC, while the marker represents the average NL2E over all DAGs in a MEC, averaged over all causal models. Since the baselines (marginal/conditional Shapley) instead produce a single Shapley vector, we then report the average NL2E over all causal models. For completeness, we also report the standard error over the causal models for the same setting in A.3.2, showing that our results are stable across the different models.

We observe that as expected the estimated causal Shapley values resulting from the oracle MEC are closest to the ground truth causal Shapley values, but they still often do not include the ground truth causal Shapley values, because while the Oracle MEC has access to the ground truth MEC, it still uses the same $n_{obs}$ finite sample data as other methods to estimate the conditional distributions. We can see that the average NL2E for all the MEC Shapley methods is still notably lower than the baselines, although the improvements are limited by the accuracy of causal discovery in finite sample cases. For the oracle MEC and the FGES MEC we can see that even the worst explanations in the range can have comparable or better performances than the baselines. As expected, the Oracle MEC method consistently has the lowest NL2E, but as expected still has a relatively large spread of the NL2E due to the variability across the DAGs in the ground truth MEC. When comparing the performance of FGES and PC, we observe that the resulting explanation quality (NL2E) does mostly, but not always correlate directly with the global graph structure similarity (SHD). For instance, in the linear setting with 16 nodes (p=16), FGES yields explanations with lower error than PC (Fig. 2a), despite having a slightly higher average SHD (Appendix Table 3).

For non-linear causal models we can see in Fig. 2b that all our methods provide better explanations on average, and that in some cases, even the worst explanation in our range is better than the baselines. Finally, we see a slight difference between Sampling and IW, with Sampling outperforming IW. Surprisingly, here the marginal Shapley values perform better relative to the true causal Shapley values than the conditional ones.

Fig. 3a shows the NL2E between each method and the ground truth causal Shapley as a function of the expected number of neighbors $d = 1, 2, 3$ for linear Gaussian models with $p = 10$ and $\bar{m} = 6$. As expected the oracle MEC Shapley is closest to the ground truth, and is not significantly affected by the expected

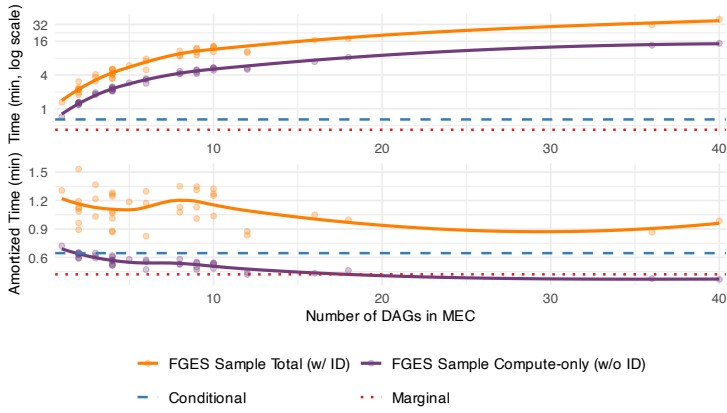

Figure 4: Breakdown of computation time for 40 SCMs (scatter points) in the linear-Gaussian regime. (top) The absolute time for our method (solid lines) compared to baselines (dashed/dotted). (bottom) The amortized time per explanation, showing that our method becomes more efficient than the conditional baseline as the number of DAGs in the MEC increases. The time measured is the computation time (total or amortized) for computing the Shapley values for 40 datapoints. We plot LOESS-smoothed curves to highlight the trend. The plots for the Oracle and PC MECs look similar.

number of neighbors, which is as we can see for $d = 3$, not the case for PC and FGES MEC, but they on average still show comparable or closer agreement than the baselines. As the expected number of neighbors $d$ increases, the baselines perform comparatively worse. The difference between Sampling and IW is again minimal. Fig. 3b shows that in the linear setting $\bar{m}$, the expected number of parents, has no strong effect on performance.

In App. A.3 Fig. 8 we show how the methods compare when we consider *only* the cases where PC and FGES output an incorrect CPDAG. We can see that overall, performance is not strongly affected. In almost all settings the average explanation is closer than the baselines We show the mean and standard errors for these results in Fig. 9. We consider $d$ and $\bar{m}$ for the non-linear case in Fig. 11, where we see that in most cases the worst explanation by both PC and FGES still outperforms the baselines. We provide ablations in terms of number of Monte Carlo samples $n_{mc}$ in Fig. 12, which do not seem to influence the results substantially. We also vary the number of samples to estimate the conditional distributions $n_{obs}$ in Fig. 13, the number of combinations used to compute the Shapley values $n_{comb}$ in Fig. 14 and the number of samples used to estimate the MEC $n_{cd}$ in Fig. 15. As expected all methods perform slightly better with more samples and higher numbers of combinations, but the differences between the baselines and the MEC Shapley methods are qualitatively similar as what presented in the previous plots. In App.A.3.10 we show an analysis of the learned MECs in all settings, reporting average sizes for PC and FGES and the quality of the CPDAGs compare to the oracle CPDAGs as measured by the Structural Hamming Distance (SHD). In Fig. 17 we show how the SHD affects the within-MEC range of NL2E. Additionally, in App. A.3.3 in the non-linear setting, we compare with conditional Shapley values with two more flexible conditional estimators (Gaussian copula and conditional inference trees) and show that the results of the baselines improve only marginally, while our methods, even with the conditional Gaussian estimators, still outperform the baselines.

Figure 4 evaluates the average time to compute the Shapley values for 40 data points, comparing the FGES Sample method against the marginal and conditional baselines as a function of the number of DAGs in the learned MEC. We report per-SCM computation times (scatter points), and overlay LOESS-smoothed curves to highlight the overall trend. The top panel illustrates the absolute computation time. As expected, our full method is more computationally expensive than the baselines, as it must compute explanations for multiple DAGs. Furthermore, it incurs a significant, but fixed, one-time overhead for the ID algorithm to identify opportunities for amortization (`Total w/ ID`). Crucially, however, we observe empirically that the total runtime scales favorably, growing sub-linearly with the MEC size, relative to a naive per-DAG iteration baseline, which would scale linearly with the amount of DAGs in the MEC. This suggests the

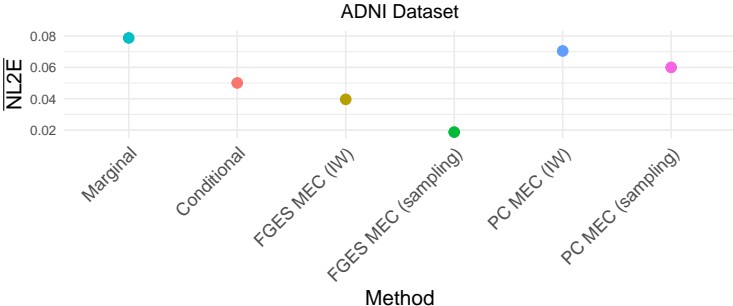

Figure 5: Average NL2E w.r.t the ground truth generated using the "gold standard" graph from Shen et al. (2020) on the ADNI dataset. Standard errors are too small to be visible.

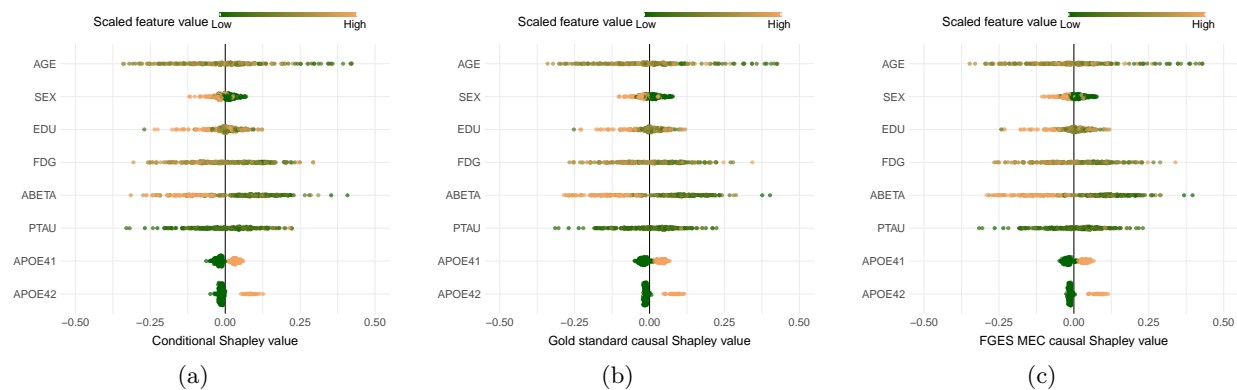

Figure 6: Sina plots for a binary classification task using XGBoost on the Alzheimer's dataset, showing the (a) conditional Shapley values, (b) causal Shapley values for the "gold standard" DAG, (c) FGES MEC Shapley values. Sina plots for these and also the other methods are in Fig. 19.

amortization identified by the ID step can materially reduce average cost per explanation as the MEC size grows, though this is an empirical observation rather than a complexity bound. The plot also decomposes the cost, revealing that while the ID algorithm is a major component, the core explanation cost (`Compute-only w/o ID`) scales even more efficiently. The bottom panel reveals the practical benefits of this structure by showing the *amortized time per DAG*. As the MEC size increases, this amortized cost continues to fall and begins to flatten, demonstrating that the marginal cost of explaining an additional causally equivalent graph becomes negligible. Even when including the initial ID overhead, the true average cost per explanation (`Amortized Total w/ ID`) consistently decreases, highlighting the power of amortizing computation.

In summary, our framework presents a computational trade-off: a higher, fixed initial cost to identify shared causal structures is exchanged for per-explanation efficiency and scalability as the causal uncertainty (i.e., MEC size) increases.

## 5.2 Real-world data

We apply our methods to two real-world settings: the Alzheimer's disease dataset obtained from the Alzheimer's Disease Neuroimaging Initiative (ADNI) database (http://adni.loni.usc.edu) and to the bike rental dataset from (Fanaee-T & Gama, 2014), both used in Heskes et al. (2020). We provide an explanation of the datasets and detailed results in App. A.4. For the Alzheimer's dataset we consider causal Shapley values based on a "gold standard graph" from expert knowledge (Shen et al., 2020) as ground truth, where we trained a prediction model to predict the probability of Alzheimer's diagnosis.

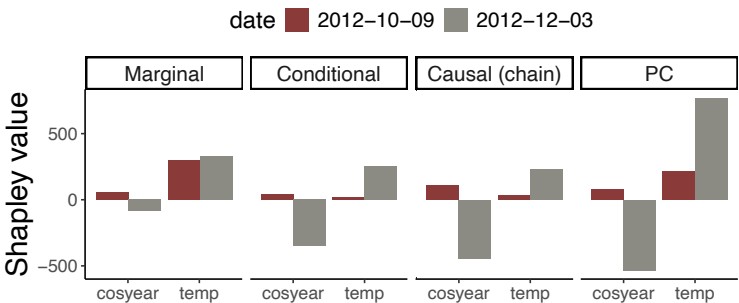

Figure 7: Bar plot for the features *cosyear* and *temp* of the bike rental dataset for two days in October and December with approximately the same temperature. We show the Causal chain Shapley values (Heskes et al., 2020), along with marginal, conditional, and PC Shapley values. FGES Shapley values are similar to PC and have been omitted.

Fig. 5 shows how the outcomes of the different algorithms compare to the ground truth Shapley values, as well as the baselines. The output of both FGES and PC is a MEC with a single graph, leading to a single causal Shapley value for each feature. We see that the FGES MEC methods are closer to the ground truth causal Shapley values than the baselines, especially when using the sampling method. The PC MEC methods are comparable to marginal and conditional Shapley values. This result highlights the critical interplay between the causal discovery and explanation stages of our pipeline. As shown in Appendix Fig. 18, the CPDAG learned by PC only learns correctly two edges, missing two edges and incorrectly orienting the edge between fudeoxyglucose (FDG) and amyloid beta (ABETA). On the other hand, the CPDAG learned by FGES correctly identifies four edges, missing only the connection between age and amyloid beta (ABETA). These errors in the graph structure likely explain why the PC MEC Shapley values are less aligned with the ground truth. This suggests that the benefit of our framework is directly tied to the ability to learn a reasonably accurate causal structure. When the discovery step is successful (as with FGES), the resulting explanations show notable improvement. We also shwo the sina plots for conditional, ground truth causal Shapley values and FGES MEC Shapley valuesin Fig. 6. The sina plots for all methods are in Fig. 19.

In Fig. 7 we look at the Shapley values for predictions of bike rental counts on two different dates with approximately equal temperature. Here the aim is a qualitative illustration, in line with Heskes et al. (2020), showing how different Shapley formulations can yield distinct explanations for comparable prediction scenarios. While marginal Shapley values provide more or less the same explanation for both days, focusing mostly on the temperature variable, we can see that causal and PC Shapley values assign credit to both season and temperature, resulting in different explanations for the two days. In App. A.4 we show additional sina plots for both datasets that show how the features get assigned different levels of importance by each method, in addition to providing more details on the real-world experiments.

## 6 Conclusion and limitations

In this paper, we combined causal discovery with causal Shapley values to provide a more nuanced explanation of predictive models, particularly when the causal graph among features is not known. Our experimental results show that our methods provide explanations that are more closely aligned with the true causal impacts of features on predictions, compared to traditional Shapley value approaches that do not incorporate causal reasoning at a limited computational cost. Our results suggest that even when the causal graph is not in the set of learned graphs, the explanations our methods output are often closer to the ground truth causal Shapley values than other methods.

While our approach marks a step forward, several limitations need to be acknowledged. First, our claims are limited to the accuracy and computational efficiency with which we can approximate the true underlying causal Shapley values. Whether causal Shapley values offer a better explanation than those provided by other methods requires interpretation, and will depend on the particular analysis use case under consideration.

Since our method returns a set of possible explanations (one per DAG in the MEC), practitioners should interpret these as reflecting causal uncertainty rather than a single definitive attribution.

A computational limitation is the scalability of our pipeline in terms of number of nodes and graph size. In particular, we are limited by the computational cost of causal discovery, which is generally an NP hard problem (Chickering et al., 2004), and even more so by Shapley estimation (which is exponential in the number of features without coalition subsampling). In terms of accuracy, we are dependent on the causal discovery methods that is employed, which can be inaccurate under wrong parametric assumptions or for small data samples. Similarly, the accuracy of both the Sampling and IW method depend on the correctness of the estimated conditional distributions. Our methods also assume causal sufficiency, i.e. no unmeasured confounders or selection bias, which may not always hold in real-world settings.

### Acknowledgments

We thank SURF for the support in using the Dutch National Supercomputer Snellius. DR and JWvdM acknowledge the support of the Bosch Center for Artificial Intelligence. SM would like to acknowledge the VIDI grant CANES (VI.Vidi.243.247).

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

# A  Appendix

## A.1  Causality background

We introduce a short background in causality. We assume that the data-generating process can be modelled using a Structural Causal Model (SCM) (Pearl, 2009; Spirtes et al., 2000), such that each variable $X_i$ for $i \in [p]$ has an assignment,

$$X_i \leftarrow f_i(\mathbf{X}_{\mathrm{Pa}(i)}, U_i), \quad i = 1, \ldots, p, \tag{10}$$

where $\mathbf{X}_{\mathrm{Pa}(i)}$ are the *parents*, i.e. the variables that cause $X_i$ directly, and $U_i$ is a noise variable, with each $U_i$ being independent of the other noise variables. This SCM induces a joint probability distribution $P(\mathbf{X}) = \prod_{i=1}^{p} P(X_i | \mathbf{X}_{\mathrm{Pa}(i)})$, as well as a graph $G$. If the graph contains no cycles, we call it a directed acyclic graph (DAG). A do-intervention, denoted by $do(\mathbf{X}_S = \mathbf{x}_S)$, is a replacement of the causal mechanism for the variables $\mathbf{X}_S$ with constant $\mathbf{x}_S$. This results in a new distribution $P(\mathbf{X}|do(\mathbf{X}_S = \mathbf{x}_S))$ that factorizes according to the truncated factorization formula:

$$P(\mathbf{X}|do(\mathbf{X}_S = \mathbf{x}_S)) = \prod_{j \notin S} p(X_j | \mathbf{X}_{\mathrm{Pa}(j)}) \delta(\mathbf{X}_S = \mathbf{x}_S), \tag{11}$$

where $\delta$ is a Dirac delta function that forces the value of $\mathbf{X}_S$ to $\mathbf{x}_S$.

## A.2  Relation between Causal and Conditional Shapley Values

In this section, we discuss the relation between conditional (Aas et al., 2021) and causal Shapley values (Heskes et al., 2020), first showing a theoretical analysis that they only coincide when there are no causal relations between the features and then providing a toy example illustrating more concretely how they differ.

### A.2.1  Theoretical analysis: conditional $\equiv$ causal Shapley values if features are independent

We provide a theoretical analysis showing that, in general, conditional and causal Shapley values are only equivalent if features are independent of each other. We start by showing how their values functions relate.

The value function for a coalition $S$ for conditional and causal Shapley values will coincide iff $P(\mathbf{X}_{\bar{S}} \mid \mathbf{X}_S = \mathbf{x}_S^*) = P(\mathbf{X}_{\bar{S}} \mid do(\mathbf{X}_S = \mathbf{x}_S^*))$. As shown by Rule 2 in do-calculus (Pearl, 2009), in our causally sufficient setting this is true if $\mathbf{X}_S \perp \mathbf{X}_{\bar{S}}$, i.e., they are d-separated (Pearl, 2009) by the empty separating set, in the causal graph $G_{\underline{\mathbf{X}}_S}$, i.e., the true causal graph after we have remove the edges outgoing of $\mathbf{X}_S$.

In other words, in this setting, the interventional and observational conditional distribution coincide iff $\mathbf{X}_S$ only contains *non-descendants* of $\mathbf{X}_{\bar{S}}$, i.e., variables that are not descendants of any of the variables $\mathbf{X}_{\bar{S}}$. This is a quite straightforward and well-known implication, but for completeness we provide a short proof here:

**Lemma.** *Let $G$ be a causally sufficient DAG over features $\mathbf{X} = \{X_1, \ldots, X_p\}$. Under the causal Markov and faithfulness assumptions, $P(\mathbf{X}_{\bar{S}} \mid \mathbf{X}_S = \mathbf{x}_S^*) = P(\mathbf{X}_{\bar{S}} \mid do(\mathbf{X}_S = \mathbf{x}_S^*))$ for any value $\mathbf{x}_S^*$ in the domain of $\mathbf{X}_S$ if a set of variables $\mathbf{X}_S \subseteq \mathbf{X}$ only contains non-descendants of the other variables $\bar{S} := \mathbf{X} \setminus \mathbf{X}_S$.*

*Proof.* Intuitively, if $X_1 \in \mathbf{X}_S$ is a descendant of $X_2 \in \mathbf{X}_{\bar{S}}$, then there exists a directed path from $X_2 \rightarrow \cdots \rightarrow X_1$, which cannot be blocked by the empty separating set, so $\mathbf{X}_S \not\perp \mathbf{X}_{\bar{S}}$ in $G_{\underline{\mathbf{X}}_S}$ and hence we cannot apply Rule 2 in do-calculus (Pearl, 2009) and these two distributions do not coincide in general. If none of the $X_1 \in \mathbf{X}_S$ are a descendant of any $X_2 \in \mathbf{X}_{\bar{S}}$, then either: (i) there are no paths between them, so they are

trivially d-separated; (ii) all paths that exist between any variable $X_1 \in \mathbf{X}_S$ and a variable $X_2 \in \mathbf{X}_{\bar{S}}$ that have an edge out of $X_1$ are cut in $G_{\underline{\mathbf{X}_S}}$, so they are all trivially blocked; (iii) all paths that exist between any variable $X_1 \in \mathbf{X}_S$ and a variable $X_2 \in \mathbf{X}_{\bar{S}}$ that have an edge into $X_1$ have a collider, i.e., $X_1 \leftarrow \cdots \rightarrow \cdots \leftarrow \ldots X_2$, since otherwise they would be directed paths from $X_2$ to $X_1$, making $X_2$ an ancestor of $X_1$, or conversely, $X_1$ a descendant of $X_2$. These paths are also then trivially blocked, since they contain a collider. If all paths between $\mathbf{X}_S$ and $\mathbf{X}_{\bar{S}}$ are blocked in $G_{\underline{\mathbf{X}_S}}$, then $\mathbf{X}_S$ and $\mathbf{X}_{\bar{S}}$ are d-separated in $G_{\underline{\mathbf{X}_S}}$. Hence by the application of Rule 2 in do-calculus (Pearl, 2009), then these two distributions coincide. $\qquad\square$

This result shows that in general the observational and interventional distribution might not coincide, unless this condition holds or for special choices of structural causal models and specific interventional values $\mathbf{x}_S^*$. We now use it to show that for a given $S \subseteq [p]$, the value functions for conditional Shapley values, which we denote as $v_{\mathrm{cond}}(S)$, and causal Shapley values, which we denote as $v_{\mathrm{causal}}(S)$ coincide if $\mathbf{X}_S$ only contains non-descendants of the other variables $\mathbf{X}_{\bar{S}}$, or if the function $f$ only depends on the variables $\mathbf{X}_S$. More formally:

**Corollary 1.** *Let $G$ be a causally sufficient DAG over features $\mathbf{X} = \{X_1, \ldots, X_p\}$. Under the causal Markov and faithfulness assumptions, for a coalition $S \subseteq [p]$, the conditional and causal value functions coincide, i.e.,*

$$v_{cond}(S) := \mathbb{E}_{P(\mathbf{X}_{\bar{S}} \mid \mathbf{X}_S = \mathbf{x}_S^*)}[f(\mathbf{X}_{\bar{S}}, \mathbf{x}_S^*)] = \mathbb{E}_{P(\mathbf{X}_{\bar{S}} \mid do(\mathbf{X}_S = \mathbf{x}_S^*))}[f(\mathbf{X}_{\bar{S}}, \mathbf{x}_S^*)] =: v_{causal}(S). \tag{12}$$

*if one of the two following conditions hold:*

1. *the variables indexed by this set $\mathbf{X}_S$ only contain non-descendants of the other variables $\mathbf{X}_{\bar{S}} := \mathbf{X} \setminus \mathbf{X}_S$. This includes trivially $S = \emptyset$ and $S = [p]$ for which this always holds.*

2. *The function $f$ depends only on the variables in $\mathbf{X}_S$ and not on any of the variables in $\mathbf{X}_{\bar{S}}$ .*

*Proof.* The second condition is trivial to prove, since the expected value of the function $f$ will collapse to $f(\mathbf{x}_S^*)$ for both conditional and causal Shapley values. We now focus on proving the first condition for functions $f$ that also depend on the other variables $\mathbf{X}_{\bar{S}}$. The value functions are expectations over $P(\mathbf{X}_{\bar{S}} \mid \mathbf{X}_S = \mathbf{x}_S^*)$ and $P(\mathbf{X}_{\bar{S}} \mid do(\mathbf{X}_S = \mathbf{x}_S^*))$, respectively, so it suffices to show these distributions are identical under this condition. As shown in Lemma A.2.1, this interventional and observational conditional distribution in general coincide if $\mathbf{X}_S$ only contains *non-descendants* of $\mathbf{X}_{\bar{S}}$. $\qquad\square$

In our setting this implies that for a feature $i$, a sufficient condition for the conditional Shapley values (Aas et al., 2021), which we denote as $\phi_{\mathrm{cond},i}$, to be identical to the causal Shapley values (Heskes et al., 2020), which we denote as $\phi_{\mathrm{causal},i}$, is that the respective value functions coincide for all sets $S \subseteq [p]$. We show that this happens if the causal graph contains no edges. While in principle, there could be other cases in which the Shapley values might coincide, despite the individual value functions being different, we consider these as pathological cases. We formalize this result as follows:

**Lemma.** *Let $G$ be a causally sufficient DAG over features $\mathbf{X} = \{X_1, \ldots, X_p\}$. Under the causal Markov and faithfulness assumptions, for a feature $i$, the conditional and causal Shapley values coincide, i.e.,*

$$\phi_{cond,i} = \phi_{causal,i}, \tag{13}$$

*if $G$ contains no edges, i.e., the variables are independent.*

*Proof.* A sufficient condition for the conditional Shapley values (Aas et al., 2021), which we denote as $\phi_{\mathrm{cond},i}$, to be identical to the causal Shapley values (Heskes et al., 2020), which we denote as $\phi_{\mathrm{causal},i}$, is that the respective value functions coincide for all sets $S \subseteq [p]$. In this setting, the second condition of Cor. 1 cannot ever hold, e.g., since this would require $f$ depends at the same time only on a variable $X_i \in \mathbf{X}$ and only on a different variable $X_j \in \mathbf{X}$, which is logically impossible. So using the first condition of Cor. 1, these results would coincide if every variable is a non-descendant of all other variables, i.e., if the causal graph contains no edges. $\qquad\square$

### A.2.2 Example with three dependent features and a linear predictor.

Heskes et al. (2020) (Section 4, Fig. 1) already provide examples of differences between marginal, conditional, asymmetric and causal Shapley values.

Here we provide a slightly different example that makes the distinction between conditional and causal Shapley values more concrete also when we have a Markov Equivalence Class. We focus on a well-known causal structure, the v-structure, for which the Markov Equivalence Class only contains one DAG, to show how this affects the MEC Shapley value methods. In this specific example, a marginal Shapley value would provide the correct output, since this example is similar to the fork example by Heskes et al. (2020), but with an additional variable. On the other hand, as discussed in detail by Heskes et al. (2020), marginal Shapley values can only attribute direct effects, which might discard the contribution of a variable that has an indirect effect, e.g., in their chain example.

While in general our method allows for MECs with multiple DAGs, we focus on this simple case to show how our output differs w.r.t. conditional Shapley values, even when we are learning a causal graph from data. Moreover, even in the case of multiple DAGs in the MEC, our approach will output a list of causal Shapley values, which would in the oracle case include the one of the ground truth DAG, while conditional Shapley values will only output a single explanation.

**Structural Causal Model.** For simplicity, we consider the following linear Gaussian SCM with independent standard Gaussian noise variables $\epsilon_1, \epsilon_2, \epsilon_3$:

$$X_1 = \epsilon_1 \tag{14}$$
$$X_2 = \epsilon_2 \tag{15}$$
$$X_3 = \alpha_1 X_1 + \alpha_2 X_2 + \epsilon_3 \tag{16}$$
$$\epsilon_1, \epsilon_2, \epsilon_3 \sim N(0,1). \tag{17}$$

This SCM results in $X_1, X_2 \sim N(0,1)$ and $X_3 \sim N(0, \alpha_1^2 + \alpha_2^2 + 1)$. For simplicity, we consider a linear prediction function that by design only uses $X_1$:

$$f(x_1, x_2, x_3) = \beta x_1. \tag{18}$$

We represent this SCM graphically in the figure below. Note that this SCM is an instance of a well-known

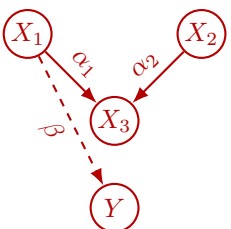

class of causal graphs, a *v-structure* (Pearl, 2009), one of the basic blocks of causal discovery. In our setting with causal sufficiency, the Markov Equivalence Class (MEC) of the v-structure only contains one DAG: the v-structure itself. So in this setting, there is only one MEC Shapley value, which coincides with the causal Shapley value with a known graph.

**Two value functions.** We compare conditional and causal value functions:

$$v_{\text{cond}}(S) := \mathbb{E}_{P(\mathbf{X}_{\bar{S}}|\mathbf{X}_S = \mathbf{x}_S^*)}\big[f(\mathbf{X}_{\bar{S}}, \mathbf{x}_S^*)\big], \tag{19}$$

$$v_{\text{causal}}(S) := \mathbb{E}_{P(\mathbf{X}_{\bar{S}}|do(\mathbf{X}_S = \mathbf{x}_S^*))}\big[f(\mathbf{X}_{\bar{S}}, \mathbf{x}_S^*)\big]. \tag{20}$$

Given one of these two value functions $v(\cdot)$, the Shapley values for features $i$ can be written as:

$$\phi_i(v) = \frac{1}{3} \sum_{S \subseteq [p] \setminus \{i\}} \binom{2}{|S|}^{-1} v(S \cup \{i\}) - v(S), \tag{21}$$

where we can already simplify some terms. In particular, we can show that regardless distribution $P \in \{P(.|\mathbf{X}_S), P(.|do(\mathbf{X}_S))\}$ and corresponding value function $v(\cdot)$:

$$v(\emptyset) = \mathbb{E}_P[f(X_1, X_2, X_3)] = \mathbb{E}_P[\beta X_1] = 0 \tag{22}$$

$$v(\{1, 2, 3\}) = \mathbb{E}_P[f(x_1, x_2, x_3)] = \mathbb{E}_P[\beta x_1] = \beta x_1 \tag{23}$$

By using Cor. 1, we can further simplify the following:

$$v_{\text{cond}}(\{1\}) = v_{\text{causal}}(\{1\}) \qquad = \beta x_1 \qquad \text{because} \quad 1 \notin Desc(\{2, 3\}) \tag{24}$$

$$v_{\text{cond}}(\{2\}) = v_{\text{causal}}(\{2\}) \qquad = 0 \qquad \text{because} \quad 2 \notin Desc(\{1, 3\}) \tag{25}$$

$$v_{\text{cond}}(\{1, 2\}) = v_{\text{causal}}(\{1, 2\}) \qquad = \beta x_1 \qquad \text{because} \quad 1, 2 \notin Desc(\{3\}) \tag{26}$$

$$v_{\text{cond}}(\{1, 3\}) = v_{\text{causal}}(\{1, 3\}) \qquad = \beta x_1 \qquad \text{because} \quad \text{f depends only on } 1. \tag{27}$$

We start by computing the part that is common to both types of Shapley values by substituting the values we have already computed:

$$\phi_1(v) = \frac{1}{3}\Big[v(\{1\}) - v(\emptyset) + \frac{1}{2}v(\{1, 2\}) - \frac{1}{2}v(\{2\}) + \frac{1}{2}v(\{1, 3\}) - \frac{1}{2}v(\{3\}) + v(\{1, 2, 3\}) - v(\{2, 3\})\Big] \tag{28}$$

$$= \frac{1}{3}\Big[\beta x_1 + \frac{\beta x_1}{2} + \frac{\beta x_1}{2} - \frac{v(\{3\})}{2} + \beta x_1 - v(\{2, 3\})\Big] \tag{29}$$

$$= \frac{1}{3}\Big[3\beta x_1 - \frac{v(\{3\})}{2} - v(\{2, 3\})\Big] \tag{30}$$

$$\phi_2(v) = \frac{1}{3}\Big[v(\{2\}) - v(\emptyset) + \frac{1}{2}v(\{1, 2\}) - \frac{1}{2}v(\{1\}) + \frac{1}{2}v(\{2, 3\}) - \frac{1}{2}v(\{3\}) + v(\{1, 2, 3\} - v(\{1, 3\})\Big] \tag{31}$$

$$= \frac{1}{6}\Big[v(\{2, 3\}) - v(\{3\})\Big] \tag{32}$$

$$\phi_3(v) = \frac{1}{3}\Big[v(\{3\}) - v(\emptyset) + \frac{1}{2}v(\{1, 3\}) - \frac{1}{2}v(\{1\}) + \frac{1}{2}v(\{2, 3\}) - \frac{1}{2}v(\{2\}) + v(\{1, 2, 3\} - v(\{1, 2\})\Big] \tag{33}$$

$$\frac{1}{3}\Big[v(\{3\}) + \frac{1}{2}v(\{2, 3\})\Big] \tag{34}$$

We now compute the Shapley value functions that differ, $v(\{3\})$ and $v(\{2, 3\})$. We start with the conditional Shapley value for feature 3:

$$v_{\text{cond}}(\{3\}) := \mathbb{E}_{P(X_1, X_2|X_3=x_3)}\big[f(X_1, X_2, x_3)\big] = \mathbb{E}_{P(X_1|X_3=x_3)}\big[f(X_1)\big] = \tag{35}$$

$$= \mathbb{E}[\beta X_1] + \beta \frac{Cov(X_1, X_3)}{Var(X_3)}(x_3 - \mathbb{E}[X_3]) \tag{36}$$

$$= 0 + \beta \frac{Cov(X_1, \alpha_1 X_1 + \alpha_2 X_2 + \epsilon_3)}{Var(X_3)}x_3 + 0 \tag{37}$$

$$= \beta \frac{\alpha_1}{\alpha_1^2 + \alpha_2^2 + 1}x_3, \tag{38}$$

where for the first equation, we first use the fact that $f$ only depends on $X_1$ and that $X_1$ and $X_2$ are independent, while then we apply the conditional expectation formula for two joint Gaussians and substitute the mean and variances we already computed. To compute the covariance, we can substitute directly the structural equation for $X_3$, where we can simplify the terms with $X_2$ and $\epsilon_3$ since they are both independent with $X_1$, resulting in $Cov(X_1, X_3) = Cov(X_1, \alpha_1 X_1) = \alpha_1$.

We compute the causal Shapley value function, which simplifies substantially since intervening on a variable that is not an ancestor of $X_1$ has no effect on its distribution, hence simplifying to:

$$v_{\text{causal}}(\{3\}) := \mathbb{E}_{P(X_1, X_2|do(X_3=x_3))}\big[f(X_1, X_2, x_3)\big] = \mathbb{E}_{P(X_1)}\big[f(X_1)\big] = 0, \tag{39}$$

which is substantially different from $v_{\text{cond}}(\{3\})$.

We focus on the conditional Shapley value for features $2, 3$. This requires a more complex computation, based on partitioning the jointly Gaussian $X_1, X_2, X_3$ in two groups: $X_1$ and $V := [X_2, X_3]^T$ with values $v = [x_2, x_3]^T$, so we can compute the conditional expectation using the standard formula for Gaussians:

$$v_{\text{cond}}(\{2,3\}) := \mathbb{E}_{P(X_1|X_2=x_2,X_3=x_3)}\big[f(X_1, x_2, x_3)\big] = \mathbb{E}_{P(X_1|X_2=x_2,X_3=x_3)}\big[f(X_1)\big] \tag{40}$$

$$= \beta\mathbb{E}[X_1] + \beta\boldsymbol{\Sigma}_{X_1V}\boldsymbol{\Sigma}_{VV}^{-1}(v - \mathbb{E}[V]) = 0 + \beta\boldsymbol{\Sigma}_{X_1V}\boldsymbol{\Sigma}_{VV}^{-1}(v - [0,0]^T) \tag{41}$$

$$= \beta\boldsymbol{\Sigma}_{X_1V}\boldsymbol{\Sigma}_{VV}^{-1}v, \tag{42}$$

where the covariance matrix $\boldsymbol{\Sigma}$ for $X_1, V$ is composed as

$$\boldsymbol{\Sigma} = \begin{bmatrix} \Sigma_{X_1X_1} & \boldsymbol{\Sigma}_{X_1V} \\ \boldsymbol{\Sigma}_{VX_1} & \boldsymbol{\Sigma}_{VV,} \end{bmatrix}.$$

We compute only the parts that we need for our value function:

$$\boldsymbol{\Sigma}_{X_1V} = \begin{bmatrix} Cov(X_1, X_2) & Cov(X_1, X_3) \end{bmatrix} = \begin{bmatrix} 0 & \alpha_1 \end{bmatrix} \tag{43}$$

$$\boldsymbol{\Sigma}_{VV} = \begin{bmatrix} Var(X_2) & Cov(X_2, X_3) \\ Cov(X_2, X_3) & Var(X_3) \end{bmatrix} = \begin{bmatrix} 1 & \alpha_2 \\ \alpha_2 & \alpha_1^2 + \alpha_2^2 + 1 \end{bmatrix}, \tag{44}$$

$$\boldsymbol{\Sigma}_{VV}^{-1} = \frac{1}{\alpha_1^2 + 1} \begin{bmatrix} \alpha_1^2 + \alpha_2^2 + 1 & -\alpha_2 \\ -\alpha_2 & 1 \end{bmatrix} \tag{45}$$

By substituting all of these in the value function formula, we get:

$$v_{\text{cond}}(\{2,3\}) = \beta \begin{bmatrix} 0 & \alpha_1 \end{bmatrix} \frac{1}{\alpha_1^2 + 1} \begin{bmatrix} \alpha_1^2 + \alpha_2^2 + 1 & -\alpha_2 \\ -\alpha_2 & 1 \end{bmatrix} \begin{bmatrix} x_2 \\ x_3 \end{bmatrix} \tag{46}$$

$$= \beta \frac{\alpha_1}{\alpha_1^2 + 1}(x_3 - \alpha_2 x_2), \tag{47}$$

We compute the causal Shapley value function for 2 and 3, which again simplifies substantially since intervening on a variable that is not an ancestor of $X_1$ has no effect on its distribution, hence simplifying to:

$$v_{\text{causal}}(\{2,3\}) := \mathbb{E}_{P(X_1|do(X_2=x_2,X_3=x_3))}\big[f(X_1, x_2, x_3)\big] = \mathbb{E}_{P(X_1)}\big[f(X_1)\big] = 0, \tag{48}$$

which is again substantially different from $v_{\text{cond}}(\{2,3\})$.

**Difference in Shapley values.** These changes in the value functions contribute to a change in the Shapley values. We use the simplified equations from before and we can now see clearly the difference in conditional (Aas et al., 2021) and causal Shapley values (Heskes et al., 2020) for feature 1:

$$\phi_{\text{cond1}} = \frac{1}{3}\big[3\beta x_1 - \frac{v_{\text{cond}}(\{3\})}{2} - v_{\text{cond}}(\{2,3\})\big]$$

$$= \beta x_1 - \frac{\beta}{3}\left[\frac{\alpha_1}{2(\alpha_1^2 + \alpha_2^2 + 1)}x_3 + \frac{\alpha_1}{\alpha_1^2 + 1}(x_3 - \alpha_2 x_2)\right]$$

$$\phi_{\text{causal1}} = \frac{1}{3}\big[\beta x_1 - v_{\text{causal}}(\{3\}) - v_{\text{causal}}(\{2,3\})\big] = \beta x_1,$$

The causal Shapley value correctly attributes the explanation to the value $x_1$ with the correct coefficient. On the other hand, the conditional Shapley value provides a biased output that can be quite far from the true causal effect, depending on the values of the datapoint and the coefficients in the SCM. Similarly, for features 2 and 3 we get:

$$\phi_{\text{cond2}} = \frac{1}{6}\big[v_{\text{cond}}(\{2,3\}) - v_{\text{cond}}(\{3\})\big] = \frac{\beta}{6}\left[\frac{\alpha_1}{\alpha_1^2 + 1}(x_3 - \alpha_2 x_2) - \frac{\alpha_1}{\alpha_1^2 + \alpha_2^2 + 1}x_3\right]$$

$$\phi_{\text{causal2}} = \frac{1}{6}\big[v_{\text{causal}}(\{2,3\}) - v_{\text{causal}}(\{3\})\big] = 0$$

$$\phi_{\text{cond3}} = \frac{1}{3}\left[v_{\text{cond}}(\{3\}) + \frac{1}{2}v_{\text{cond}}(\{2,3\})\right] = \frac{\beta}{3}\left[\frac{\alpha_1}{\alpha_1^2 + \alpha_2^2 + 1}x_3 + \frac{\alpha_1}{2(\alpha_1^2 + 1)}(x_3 - \alpha_2 x_2)\right]$$

$$\phi_{\text{causal3}} = \frac{1}{3}\left[v_{\text{cond}}(\{3\}) + \frac{1}{2}v_{\text{cond}}(\{2,3\})\right] = 0$$

Neither of these two features has any effect on the prediction, so the causal Shapley value provides the correct explanation, while the conditional Shapley value provides a biased explanation due to conditioning also on variables that are descendants of $X_1$. In this specific example, a marginal Shapley value would provide the correct output, since this is a variation of the fork example in Heskes et al. (2020) with an additional variable. On the other hand, marginal Shapley values disregard indirect effects on the prediction, which might be an issue in many settings in which the variable that is used in the prediction is just a proxy of the actual variable that influences the prediction, e.g., a ZIP code in credit card approval system, or other types of *unresolved discrimination*, as discussed by Frye et al. (2020).

### A.3    Complete experiments - synthetic data

Below we present the full set of results for all experiments performed. We investigate what happens when we only consider CPDAGs that were learned incorrectly, the range provided by each MEC Shapley, and ablations for the hyperparameters $n_{\mathrm{mc}}$, $n_{\mathrm{obs}}$, $n_{\mathrm{cd}}$, $n_{\mathrm{comb}}$.

#### A.3.1    Performance on incorrectly learned CPDAGS

In Fig. 8 we report the same metrics as in Fig. 2 in the main text, but now including only models for which neither PC and FGES learned the correct CPDAG, i.e. the oracle CPDAG. For each setting we report on how many models out of the original 40 are learned incorrectly: for Fig. 8a the number of models analyzed are 12, 18, and 26 for $p = 5, 10, 15$. For Fig. 8b, the numbers are $16, 32$, and $36$ for $p = 5, 10, 15$. For Fig. 8c, we have $5, 18, 30$ models for $d = 1, 2, 3$, Overall this ranges from a third for the simpler settings (linear, small number of nodes or sparse) to half for the default setting and almost all for the non-linear case. We see that in almost all cases results are not strongly affected by excluding the models that were learned perfectly. In almost all settings the average explanation is better than the marginal/conditional Shapley values, and in many cases even the worst explanation is comparable to, and sometimes better than, the baselines.

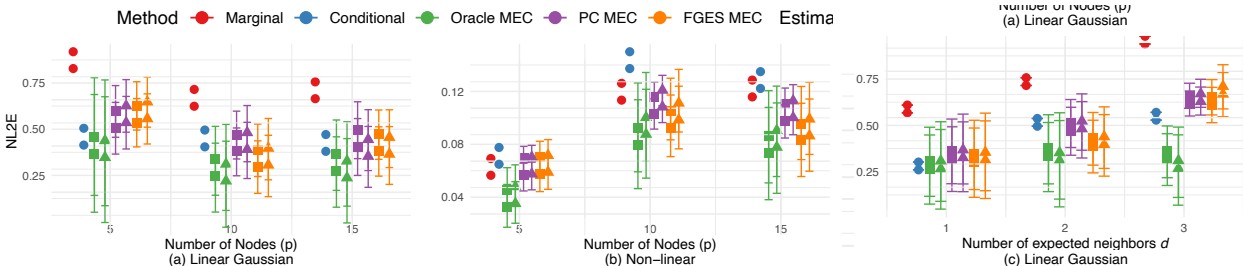

Figure 8: Normalized L2 error between the ground truth causal Shapley value and the Shapley values returned by each method for different parameters, where only models are included where *both* PC and FGES learned an incorrect CPDAG, i.e. a CPDAG not equivalent to the oracle CPDAG. The amount of models included is (a) $12, 18, 26$ for $p = 5, 10, 15$ in the linear Gaussian setting, (b) $16, 32, 36$ for $p = 5, 10, 15$ in the non-linear setting, (c) $5, 18, 30$ for $d = 1, 2, 3$ in the linear Gaussian setting.

#### A.3.2    Mean and standard error plots for Fig. 2

For completeness, in Fig. 9 we report the mean and standard error of the NL2E, averaged over the DAGs in the MEC and over the $n_{\mathrm{scm}} = 40$ causal models. In particular, the standard error is computed over $n_{\mathrm{scm}} = 40$ causal models, after having averaged over DAGs. We report for (a) the linear Gaussian setting where we vary $p$, (b) the non-linear setting with varying $p$, (c) the linear Gaussian setting with varying $d$. These results show the stability of the evaluation over the different causal models.

#### A.3.3    More flexible conditional estimators

To assess whether the performance gap between our method and the baselines is simply due to the misspecification of the Gaussian conditional estimator in the non-linear setting, we conduct an additional ablation study.

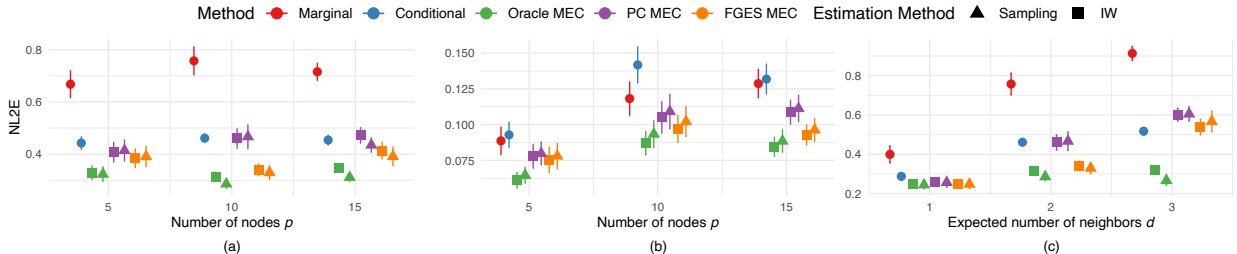

Figure 9: Mean and standard error for NL2E between the ground truth causal Shapley value and the Shapley values returned by each method for different parameters and settings: (a) linear Gaussian setting with varying number of features $p$, (b) nonlinear Gaussian setting with varying number of features $p$, (c) linear Gaussian setting with varying number of expected neighbors $d$. For all MEC Shapley methods we plot the average NL2E over all the explanations, with error bars representing the SE over $n_{\mathrm{scm}} = 40$ causal models.

We extend the Conditional Shapley baseline by employing two more flexible, non-parametric conditional density estimators provided by the `shapr` package: Gaussian Copula and Conditional Inference Trees (`ctree`). Figure 10 reports these results. Note that while we enhance the baselines with these flexible models to capture non-linear dependencies, the MEC Shapley variants continue to use the simpler Gaussian conditional modeling. This provides a conservative comparison, testing whether causal structure learning (even with misspecified parametric assumptions) yields better explanations than sophisticated conditional modeling that ignores causality.

Because the true data-generating mechanism is non-linear, Gaussian conditionals are misspecified. Nevertheless, as shown in Fig. 10, MEC Shapley values remain systematically closer to the ground-truth *causal* Shapley values (computed from the true interventional distributions) than conditional Shapley values. Using more flexible conditional estimators narrows the gap to the causal target for the conditional baseline, but it does not remove the advantage of MEC Shapley: on average, MEC Shapley values are still closer to the causal Shapley values, even when MEC Shapley is computed under the misspecified Gaussian conditional model.

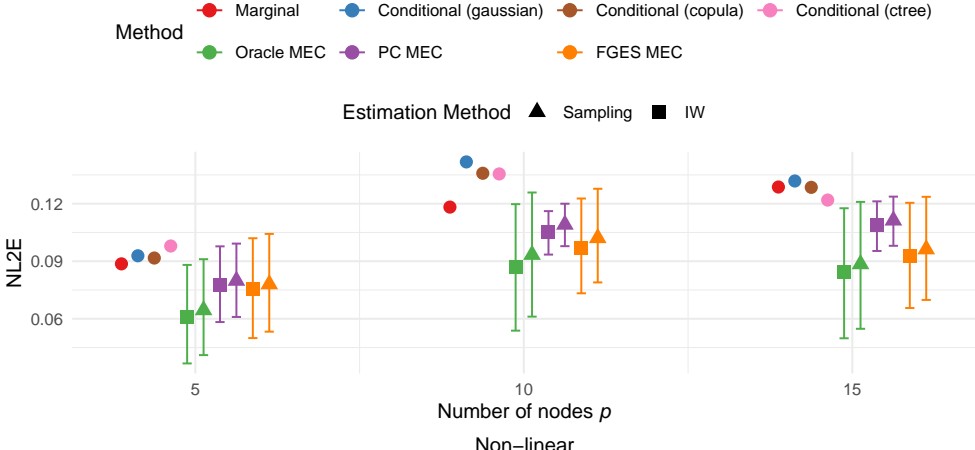

Figure 10: Ablation study comparing Conditional Shapley values with flexible non-linear estimators against our MEC Shapley framework in the non-linear setting. We report the Normalized L2 Error (NL2E) for Conditional Shapley values estimated via Gaussian assumptions (standard), Gaussian Copula, and Conditional Inference Trees (`ctree`). Even when the Conditional baseline is enhanced with flexible estimators to capture non-linearities, the MEC Shapley methods (using standard Gaussian conditionals) consistently provide explanations closer to the ground-truth causal Shapley values.

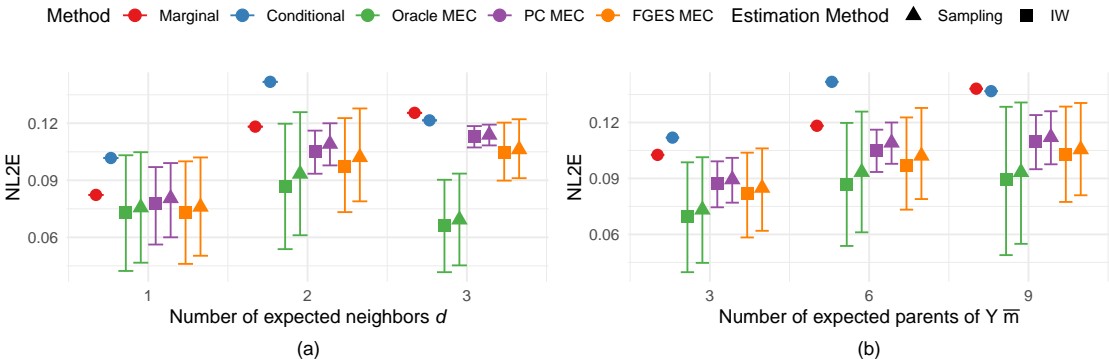

Figure 11: $NL2E$ as a function of (a) the expected number of neighbors of each node $d$ and (b) the expected number of parents of $Y$ in the non-linear setting.

### A.3.4 Varying the amount of neighbors and parents

In Fig. 11 we vary (a) the expected number of neighbors for each node $d$ and (b) the number of expected parents of Y $\bar{m}$, both in the non-linear setting. We can see that the number of expected neighbors has a strong influence on causal discovery performance in the non-linear setting, as the MEC Shapley values for PC get progressively worse. The difference between the range resulting from the oracle MEC and from the PC MEC shows that the true graph is on average not contained in the MEC, and the best graph on average leads to explanations deviating far from the one of the true graph. Increasing the expected number of parents of Y $\bar{m}$ seems to affect all methods equally, with the results getting worse as $\bar{m}$ increases. Here we fix the amount of nodes $d = 10$. We use the default settings $n_{\mathrm{cd}} = n_{\mathrm{mc}} = n_{\mathrm{obs}} = 1000$, with $n_{\mathrm{mc}} = 4000$ for the ground truth causal Shapley values.

### A.3.5 Amount of Monte Carlo samples

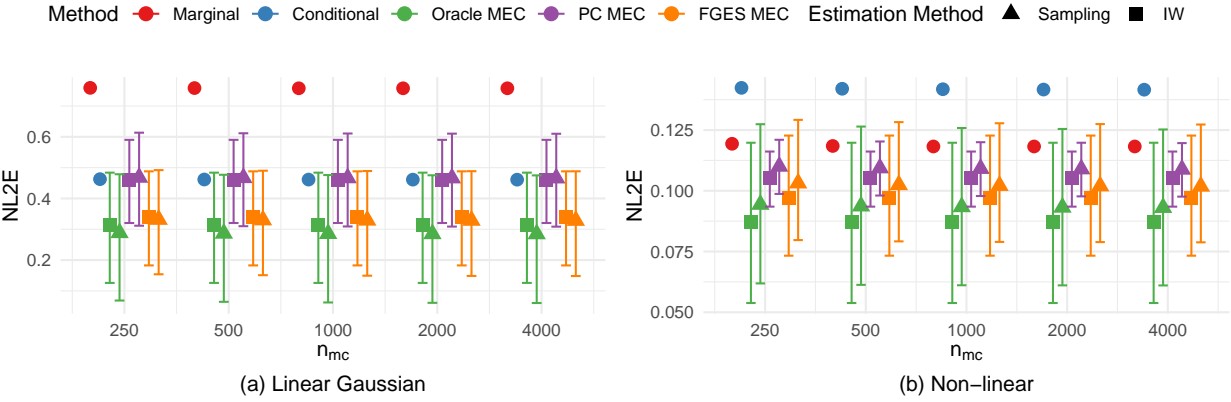

Figure 12: $NL2E$ between the ground truth causal Shapley values and other methods in (a) the linear Gaussian setting and (b) non-linear setting as a function of number of Monte Carlo samples $n_{\mathrm{mc}}$ used to estimate each expectation.

In Fig. 12 we perform an ablation where we vary $n_{\mathrm{mc}}$, the amount of Monte Carlo samples used to estimate expectations for (a) the linear Gaussian setting and (b) the non-linear setting. We can observe that the results in terms of error relative to the ground truth are, after averaging over all datapoints, not significantly influenced by the number of Monte Carlo samples $n_{\mathrm{mc}}$. Here we fix $n_{\mathrm{obs}} = 1000$. For the ground truth causal Shapley values we used $n_{\mathrm{mc}} = 4000$. In this ablation the graphs contain $p = 10$ nodes with $d = 2$ and $\bar{m} = 6$. We use the full set of combinations $S$.

### A.3.6 Amount of data for estimation of distributions

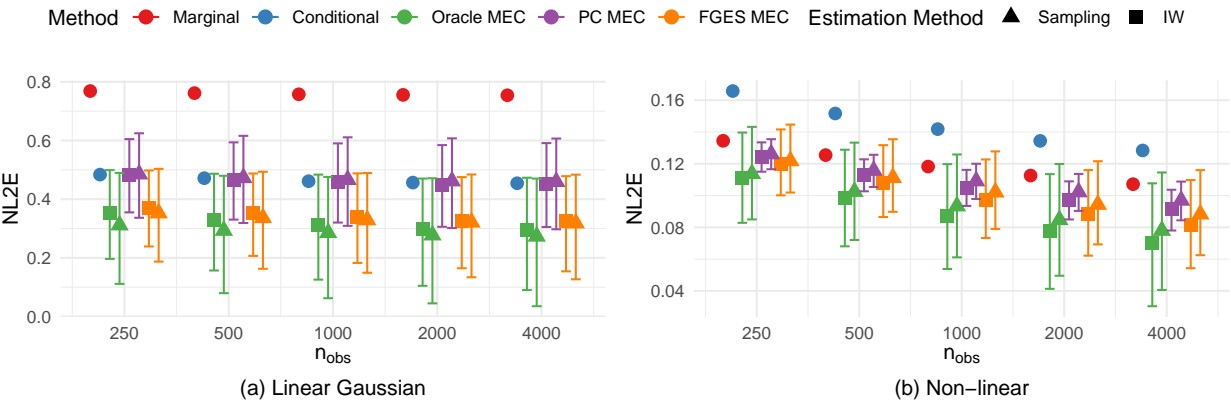

(a) Linear Gaussian

(b) Non–linear

Figure 13: $NL2E$ between the ground truth causal Shapley values and other methods in (a) the linear Gaussian setting and (b) non-linear setting as a function of number of observational samples $n_{\mathrm{obs}}$ used to estimate each marginal or conditional distribution.

Fig. 13 shows the average $NL2E$ as a function of $n_{\mathrm{obs}}$, the number of observational samples used to estimate marginal and conditional distributions, for (a) the linear Gaussian setting and (b) the non-linear setting. Here we can see that the amount of samples used to estimate distributions affects all methods equally. A low amount of samples increases the error for all explanations, but we can see that as $n_{\mathrm{obs}}$ increases, this effect starts to diminish. Interestingly, this holds also for the non-linear case, where the parametric assumptions for the conditional and MEC Shapley values fails to hold.

### A.3.7 Amount of combinations

In Fig. 14 we investigate what occurs when we use only a subset of $[p]$. Unsurprisingly, the error relative to the ground truth decreases as we use more combinations. We can see that the differences between methods stay consistent, with the average error decreasing with the amount of combinations. Here we also include the ground truth computed using only a subset of the power set of $[p]$, denoted *Ground truth (limited)*, which uses the same number of combinations $n_{comb}$ as the other methods. We can see that the error between the methods and the ground truth computed using a lower number of combinations stays consistent, which is important given that we do not use the full power set of subsets for the results on 15 nodes.

### A.3.8 Amount of data for causal discovery

In Fig. 15 we vary the amount of samples available for causal discovery for the linear Gaussian setting (a) and the non-linear setting (b). We can see that as the number of samples grows, the difference between the Oracle MEC and PC and FGES decreases, explained by the fact that the outcome of the causal discovery algorithms gets closer to the Oracle MEC.

### A.3.9 Effect of number MC samples on computation time

In Fig. 16 we plot the computation time for a single datapoint for each method. We show how $n_{\mathrm{mc}}$ affects computation time, where we report both the time taken including the identification of equivalent interventional distributions and the time taken just for computation of the Shapley values denoted as (`w/o id`). While the time taken to identify equivalent distributions is significant, once performed it markedly decreases computation time. The IW method also speeds up computation time. To compare MECs across graphs, we divide total computation time by the amount of graphs in each MEC. As expected the higher the $n_{mc}$, the higher the average running time for the MEC Shapley methods and the Conditional Shapley values, with the running time for higher numbers of MC samples comparable between Conditional Shapley and the MEC methods.

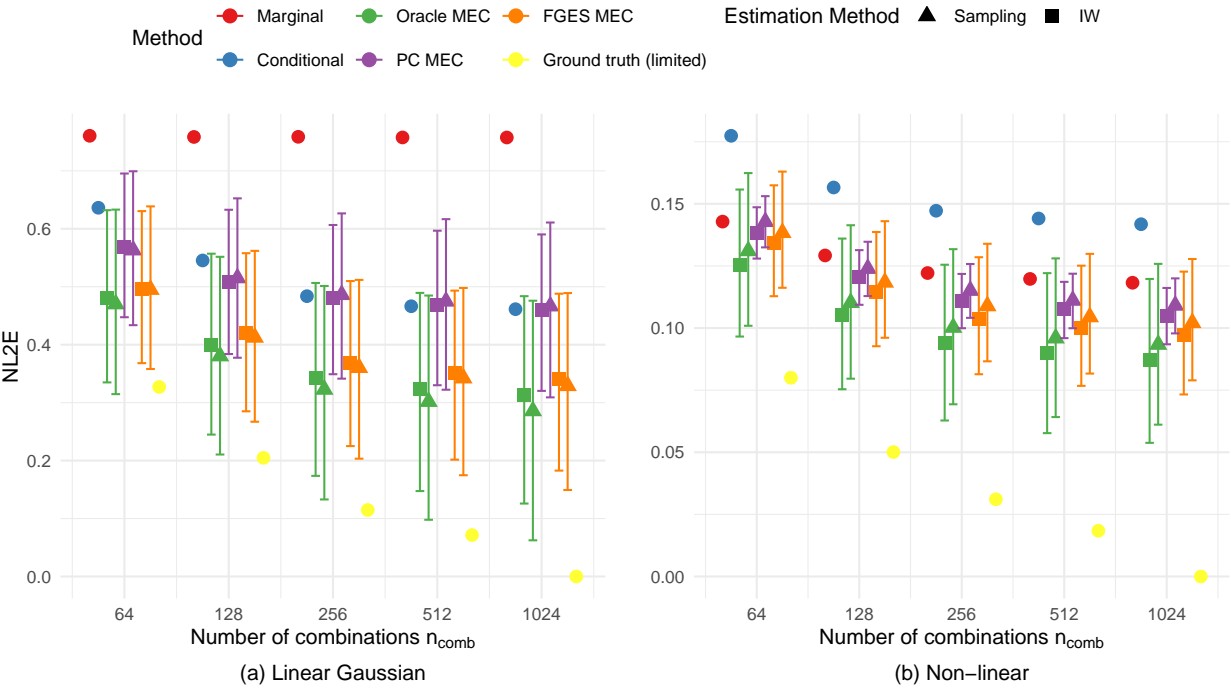

Figure 14: *NL2E* between the ground truth causal Shapley values and other methods in (a) the linear Gaussian setting and (b) non-linear setting as a function of number of combinations $n_{\text{comb}}$ used to estimate each Shapley value.

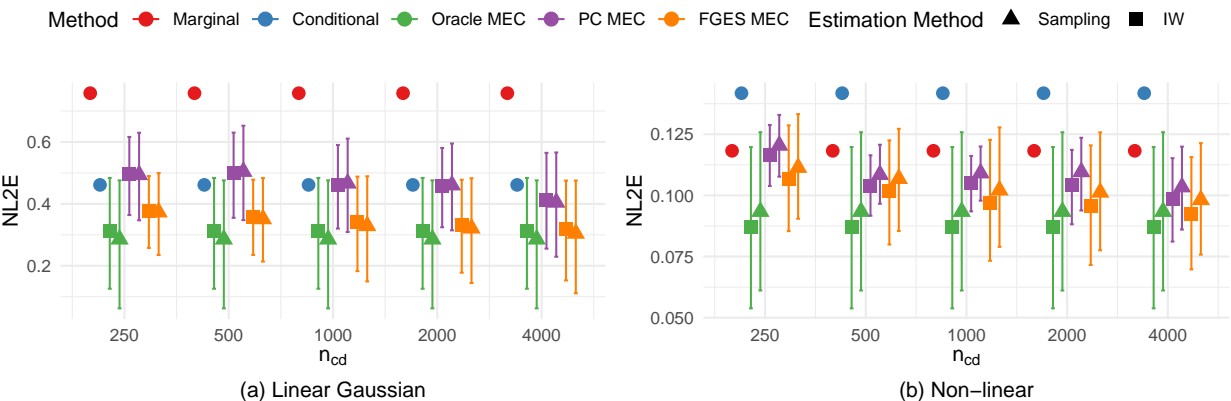

Figure 15: *NL2E* between the ground truth causal Shapley values and other methods in (a) the linear Gaussian setting and (b) non-linear setting as a function of number of causal discovery samples $n_{\text{cd}}$ used to estimate each MEC. Marginal, conditional and Oracle Shapley values are not affected by this parameter.

### A.3.10 Analysis of MECs

In Table 1 we report the average number of DAGs in each MEC as well as the standard deviation for the linear Gaussian and non-linear setting, where we vary the number of nodes $p$. In Table 2 we report the same, now varying $d$, the expected number of neighbors for each node. We do not report these tables for the expected number of parents of $Y$ $\bar{m}$, as $Y$ is not included in the causal discovery phase and as such this parameter does not influence the size of each MEC. We can see that the size of the MECs varies substantially across the different models even within the same setting.

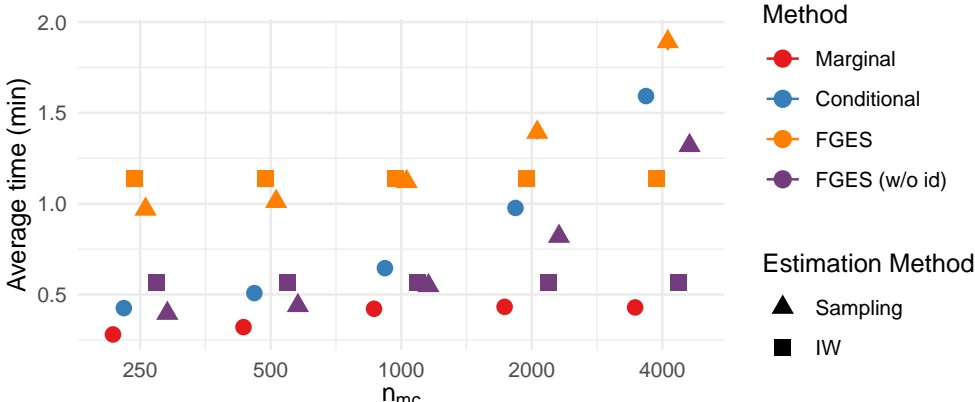

Figure 16: The average runtime for the linear Gaussian setting as a function of the number of MC samples $n_{\mathrm{mc}}$.

| Nr. of Nodes $p$ | Type | MEC size |
|---|---|---|
| 6 | FGES | $3.00 \pm 2.03$ |
| 6 | Oracle | $4.9 \pm 5.33$ |
| 6 | PC | $3.15 \pm 2.25$ |
| 11 | FGES | $7.82 \pm 8.08$ |
| 11 | Oracle | $10.2 \pm 13.0$ |
| 11 | PC | $7.42 \pm 8.20$ |
| 16 | FGES | $15.2 \pm 20.3$ |
| 16 | Oracle | $17.9 \pm 14.8$ |
| 16 | PC | $13.9 \pm 10.5$ |

| Nr. of Nodes $p$ | Type | MEC size |
|---|---|---|
| 6 | FGES | $4.55 \pm 5.10$ |
| 6 | Oracle | $4.9 \pm 5.33$ |
| 6 | PC | $2.7 \pm 2.40$ |
| 11 | FGES | $5.5 \pm 4.11$ |
| 11 | Oracle | $10.2 \pm 13.0$ |
| 11 | PC | $3.18 \pm 4.25$ |
| 16 | FGES | $11.2 \pm 12.0$ |
| 16 | Oracle | $17.9 \pm 14.8$ |
| 16 | PC | $4.97 \pm 5.48$ |

Table 1: Tables reporting MEC sizes for the linear Gaussian setting (left) and non-linear setting (right) where the number of nodes $p$ varies. We report mean and standard deviation for each setting for the Oracle, FGES, and PC MECs.

In Tables 3 and 4 we report the average Structural Hamming Distance (SHD) for each learned CPDAG with respect to the Oracle CPDAG as well as the standard deviation for the linear Gaussian and non-linear setting. In Table 3 we report these values for varying numbers of nodes $d$, and in Table 4 we vary the expected number of neighbors $\bar{m}$ for each node.

Fig. 17 relates structural discovery error to explanation error for the PC-based sampling variant for $n_{scm} = 40$ causal models. On the x-axis we report the SHD between the learned CPDAG and the oracle CPDAG for each run; on the y-axis we report NL2E. The labels on each point represent the number of CPDAGs with that SHD. For a fixed learned CPDAG, our method produces one explanation per DAG in its Markov equivalence class (MEC). Accordingly, for each run we compute NL2E for every DAG in the learned MEC and summarize

| Nr. of Neighbors $d$ | Type | MEC Size |
|:---:|:---:|:---:|
| 1 | FGES | $7.35 \pm 5.26$ |
| 1 | Oracle | $7.15 \pm 5.30$ |
| 1 | PC | $6.92 \pm 5.23$ |
| 2 | FGES | $7.82 \pm 8.08$ |
| 2 | Oracle | $10.2 \pm 13.0$ |
| 2 | PC | $7.42 \pm 8.20$ |
| 3 | FGES | $6.72 \pm 13.0$ |
| 3 | Oracle | $8.82 \pm 14.2$ |
| 3 | PC | $3.92 \pm 2.67$ |

| Nr. of Neighbors $d$ | Type | MEC Size |
|:---:|:---:|:---:|
| 1 | FGES | $6.78 \pm 4.10$ |
| 1 | Oracle | $7.15 \pm 5.30$ |
| 1 | PC | $5.12 \pm 5.41$ |
| 2 | FGES | $5.5 \pm 4.11$ |
| 2 | Oracle | $10.2 \pm 13.0$ |
| 2 | PC | $3.18 \pm 4.25$ |
| 3 | FGES | $6.3 \pm 8.66$ |
| 3 | Oracle | $8.82 \pm 14.2$ |
| 3 | PC | $2.55 \pm 3.52$ |

Table 2: Tables of MEC sizes for the linear Gaussian setting (left) and non-linear setting (right) where the number of neighbors $d$ varies. We report mean and standard deviation for each setting for the Oracle, FGES, and PC MECs.

| Nr. of Nodes $p$ | Type | SHD |
|:---:|:---:|:---:|
| 6 | FGES | $1.73 \pm 2.75$ |
| 6 | PC | $1.58 \pm 2.10$ |
| 11 | FGES | $2.00 \pm 3.66$ |
| 11 | PC | $4.03 \pm 3.50$ |
| 16 | FGES | $4.72 \pm 7.17$ |
| 16 | PC | $4.62 \pm 4.26$ |

| Nr. of Nodes $p$ | Type | SHD |
|:---:|:---:|:---:|
| 6 | FGES | $1.58 \pm 2.04$ |
| 6 | PC | $2.4 \pm 2.07$ |
| 11 | FGES | $3.85 \pm 3.08$ |
| 11 | PC | $5.65 \pm 3.04$ |
| 16 | FGES | $5.65 \pm 4.41$ |
| 16 | PC | $8.48 \pm 3.72$ |

Table 3: Tables of SHD (Structural Hamming Distance) with respect to the Oracle CPDAG for linear Gaussian setting (left) and non-linear setting (right). Mean and standard deviations are reported for FGES and PC CPDAGs across varying numbers of nodes $p$.

the resulting set by its within-MEC minimum (best-case), mean (average-case), and maximum (worst-case). The three curves plot these summaries after averaging over runs that attain the same CPDAG-SHD value, and the shaded band spans the corresponding best–worst envelope. Overall, the mean NL2E increases with CPDAG-SHD, indicating that larger discovery errors tend to degrade explanation quality. When the learned MEC contains only a single DAG, the best/mean/worst summaries coincide and the envelope collapses.

Interestingly, the NL2E spread across the DAGs in each MEC does not seem to increase predictably with higher SHD. One explanation is that the spread is completely due to the size of the MEC, which is known to be asymptotically constant and empirically on average containing 4 DAGs (Gillispie & Perlman, 2001).

## A.4 Complete experimental results - real world data

### A.4.1 Alzheimer's data set

Following Heskes et al. (2020), we consider the Alzheimer's disease data set obtained from the Alzheimer's Disease Neuroimaging Initiative (ADNI) database (http://adni.loni.usc.edu), which focuses on identifying biomarkers for early Alzheimer's detection and progression tracking. Our analysis incorporates the same features as in (Shen et al., 2020; Heskes et al., 2020), age ($AGE$), education level ($EDU$), gender ($SEX$), amyloid beta ($ABETA$), fudeoxyglucose ($FDG$), phosphorylated tau ($PTAU$), and the number of apolipoprotein alleles ($APOE41$, $APOE42$), for a total of 8 features). As ground truth for the causal discovery we employ the gold standard causal graph from Shen et al. (2020), which includes dummy-encoded variables for the apolipoprotein alleles ($APOE41$ and $APOE42$), and which we show in Fig. 18 (left). For causal discovery we discretized the continuous variables with a maximum of 4 values for each variable using the Freedman-Diaconis rule. Values that are encoded as being larger or smaller than a cutoff value, such as $ABETA > 1700$, are re-encoded at their cutoff value. Like (Heskes et al., 2020), we group together patients with mild cognitive

| Nr. of Neighbors $d$ | Type | SHD | Nr. of Neighbors $d$ | Type | SHD |
|---|---|---|---|---|---|
| 1 | FGES | $0.6 \pm 0.98$ | 1 | FGES | $1.27 \pm 1.65$ |
| 1 | PC | $0.7 \pm 1.12$ | 1 | PC | $2.6 \pm 2.53$ |
| 2 | FGES | $2.0 \pm 3.66$ | 2 | FGES | $3.85 \pm 3.08$ |
| 2 | PC | $4. \pm 3.50$ | 2 | PC | $5.65 \pm 3.04$ |
| 3 | FGES | $9.5 \pm 9.46$ | 3 | FGES | $7.38 \pm 5.29$ |
| 3 | PC | $9.3 \pm 5.16$ | 3 | PC | $9.1 \pm 3.74$ |

Table 4: Tables of SHD (Structural Hamming Distance) with respect to the Oracle CPDAG for linear Gaussian setting (left) and non-linear setting (right). Mean and standard deviations are reported for FGES and PC CPDAGs across varying expected numbers of neighbors per node $d$.

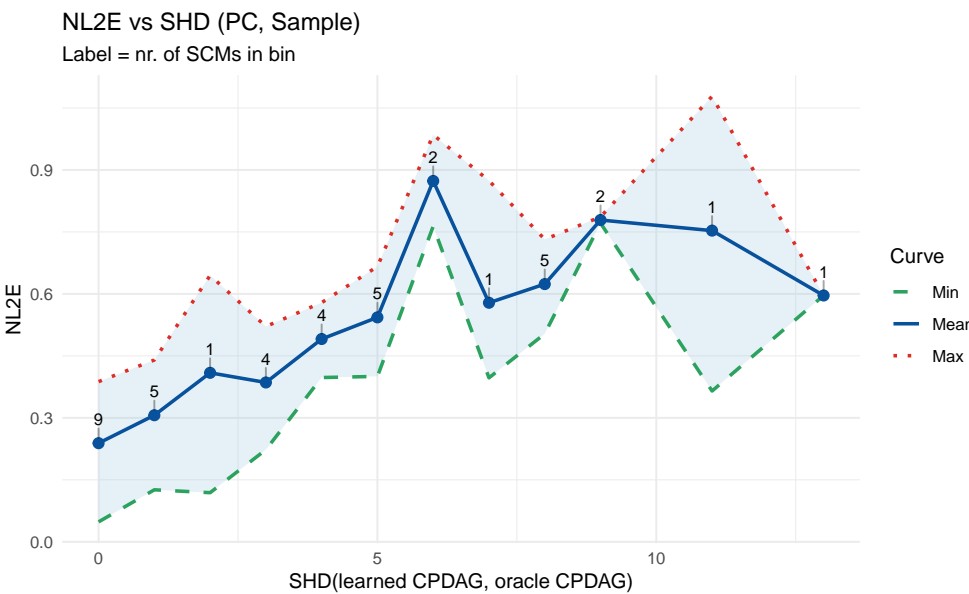

Figure 17: NL2E vs. CPDAG-SHD (PC, sampling, $n_{scm} = 40$) for $p = 10, d = 2$, with linear-Gaussian relations. On average the causal graphs have 20 edges. For each run, SHD is computed between the learned and oracle CPDAGs. We evaluate NL2E for all DAGs in the learned MEC and report the within-MEC min/mean/max (green/blue/red). Curves average these summaries over learned CPDAGs with the same SHD (the labels on each point show how many causal models have this SHD); the shaded band spans the averaged min–max range.

impairment and Alzheimer's disease, dropping patients with the diagnosis of significant memory concerns. Rows containing missing values were removed. We are left with a total of $N = 1500$ datapoints.

After these preprocessing steps, we apply the PC algorithm on a random subset of 80% of the data ($0.8 \times 1500 = 1200$ datapoints) with the G-square test with $\alpha = 0.05$, and the FGES algorithm using the discrete BIC score with penalty discount $\lambda = 1$. We prohibit edges from biomarkers into demographic variables following (Shen et al., 2020). The TETRAD library was used to run both algorithms (Ramsey et al., 2018). In Fig. 18 on the right we show the learned CPDAGs, with blue edges identified by both FGES and PC, green edges exclusively by FGES, and red edges only by PC. The blue and green edges agree with the gold standard graph, whereas the red edge ($FDG \rightarrow ABETA$) is oriented incorrectly. Compared to FGES, the CPDAG learned by PC is missing the edge from $ABETA$ to $PTAU$ and it inverts the edge from $ABETA$ to $FDG$. Neither of the methods include the edge $AGE \rightarrow ABETA$. The two MECs represented by the CPDAGs each contain only a single DAG.

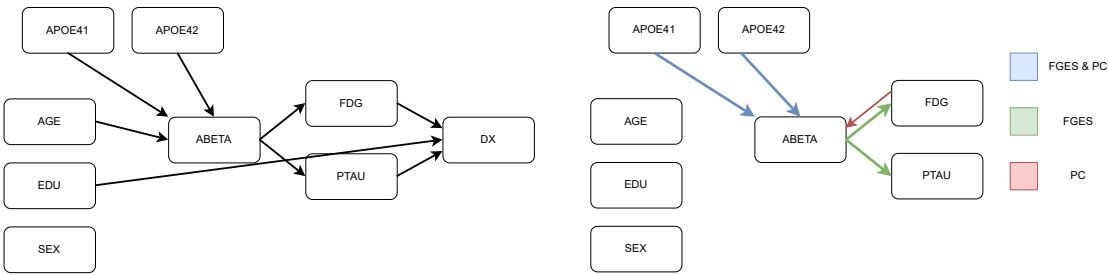

Figure 18: Comparison of gold standard graph recreated from Shen et al. (2020) (left) with CPDAGs discovered by FGES and PC algorithms (right). Edges are color-coded to indicate discovery: blue for edges identified by both FGES and PC, green exclusively by FGES, and red only by PC. The blue and green edges agree with the gold standard graph, whereas the red edge is oriented incorrectly.

Like (Heskes et al., 2020), we consider a binary classification task, grouping together patients with mild cognitive impairment and Alzheimer's disease, dropping patients with the diagnosis of significant memory concerns. We train an XGBoost model (Chen & Guestrin, 2016) for 100 rounds on the subset used for causal discovery ($0.8 \times 1500 = 1200$ datapoints), with the other 20% unseen ($0.2 \times 1500 = 300$ datapoints) by the prediction model and used for evaluation. In the sina plots in Fig. 19 we show for each feature the marginal, conditional, gold-standard causal, FGES, and PC Shapley values, respectively. We can see that FGES and ground truth Causal Shapley values both capture the indirect effect of APOE41 on the prediction, while the marginal Shapley values do not, as well as attributing a larger importance to APOE42.

**Acknowledgments.**   ADNI data collection and sharing was funded by the Alzheimer's Disease Neuroimaging Initiative (ADNI) (National Institutes of Health Grant U01 AG024904) and DOD ADNI (Department of Defense award number W81XWH-12-2-0012). See http://adni.loni.usc.edu/wp-content/themes/freshnews-dev-v2/documents/policy/ADNI_Data_Use_Agreement.pdf, section 12, for further contributions and http://adni.loni.usc.edu/wp-content/uploads/how_to_apply/ADNI_Acknowledgement_List.pdf for a complete listing of ADNI investigators.

### A.4.2   Bike rental dataset

We consider the bike rental dataset of Fanaee-T & Gama (2014), with the same features and preprocessing as used by Heskes et al. (2020): the number of days since January 2011 (*trend*), two cyclical variables to represent season (*cosyear, sinyear*), the temperature (*temp*), feeling temperature (*atemp*), wind speed (*windspeed*), and humidity (*hum*), for a total of 7 continuous features. We use the partial order ({*trend*},{*cosyear, sinyear*}, {all weather variables}) from Heskes et al. (2020) as background information for the PC and FGES algorithm, both ran using Tetrad. For PC we apply Fisher's Z test with $\alpha = 0.05$, and for FGES the BIC criterion with $\lambda = 1$. The resulting CPDAGs are shown in Fig. 20. In this dataset we do not have the ground truth causal graph and moreover, there are likely many latent confounders, so we can only reason qualitatively. The CPDAG discovered by PC is sparser than the one by FGES, missing the link between trend and temperature, as well as temperature and windspeed. Moreover the two CPDAGs disagree on the direction of the edge between windspeed and the feeling temperature *atemp*. The CPDAG discovered by FGES contains a single undirected edge between humidity and windspeed, while this edge is oriented *windspeed $\rightarrow$ hum* in the CPDAG discovered by PC. In both graphs, some of the edges that we intuitively expect are missing, e.g., *trend $\rightarrow$ cosyear*, possibly because *sinyear* and *cosyear* are deterministic functions of the trend.

We split the data (total number of $N = 730$ datapoints) into 80% ($0.8 \times 730 = 584$ datapoints) training data and 20% ($0.2 \times 730 = 146$ datapoints) test data, and trained an XGBoost model for 100 rounds, predicting the amount of bike rentals based on the previously mentioned features. In Fig 21 we show sina plots for the marginal Shapley values, the conditional Shapley values and the PC Shapley values (right). We see that the PC Shapley values better capture the contribution of cosyear than marginal and conditional Shapley values, which represents the seasonality of the bike rental (i.e. higher bike rental in summer than in winter).

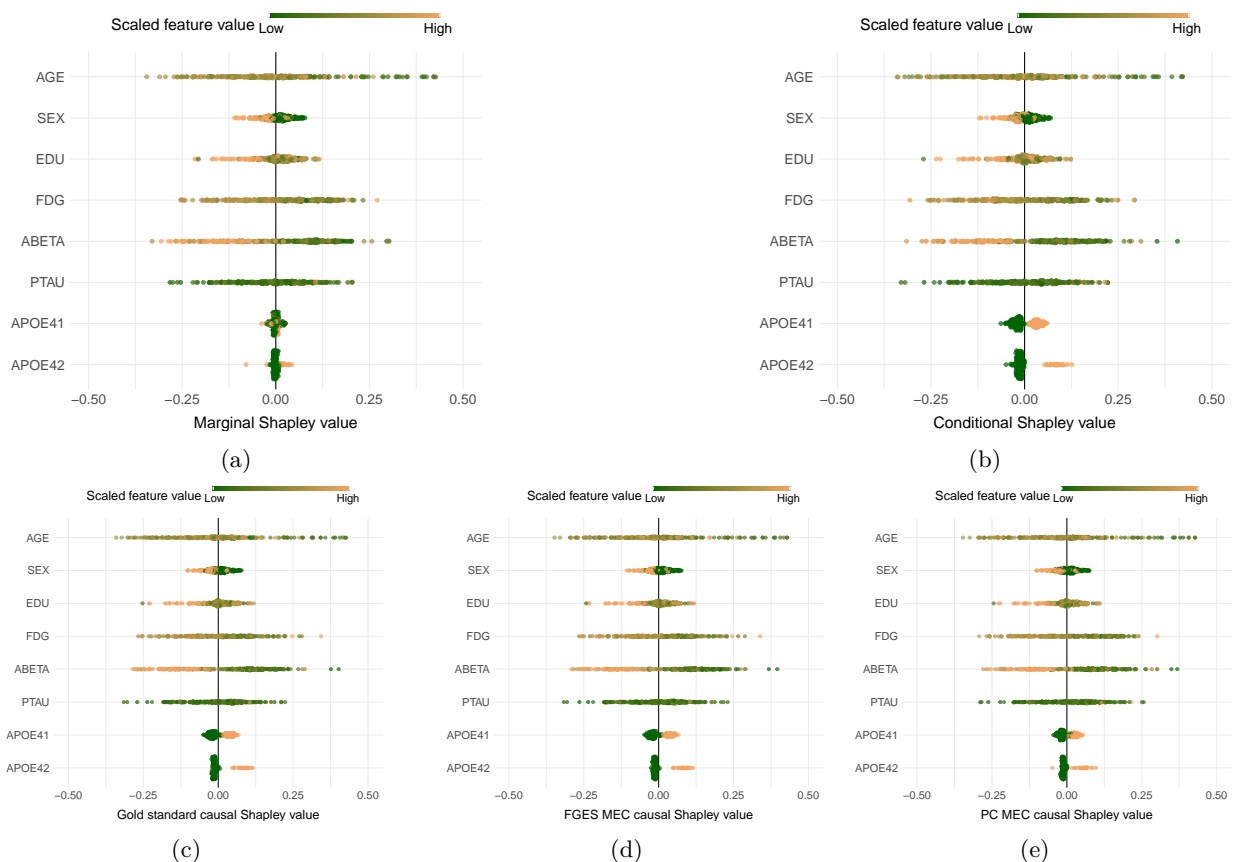

Figure 19: Sina plots for a binary classification task using XGBoost on the Alzheimer's dataset. We plot (a) marginal, (b) conditional, (c) the causal Shapley values for the "gold standard" DAG, (d) FGES MEC Shapley values and (e) PC MEC Shapley values.

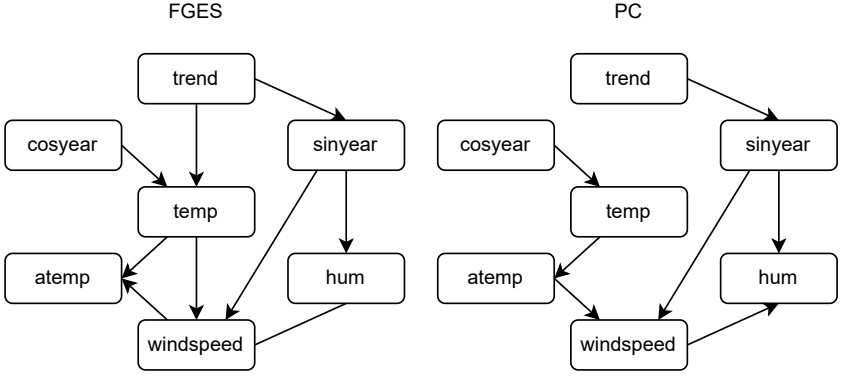

Figure 20: The learned CPDAGs on the bike rental dataset for FGES (left) and the PC algorithm (right).

The figure also shows that there is little difference in the attribution between temperature and perceived temperature when using conditional Shapley values, unlike with PC Shapley values where the difference is more noticeable. In Fig. 22 we show the sina plots for the two DAGs in the CPDAG discovered by FGES. We can observe that when windspeed causes humidity (a), the Shapley values for windspeed are slightly more spread out than when humidity causes windspeed (b). Compared to both FGES plots, in the sina plot of PC

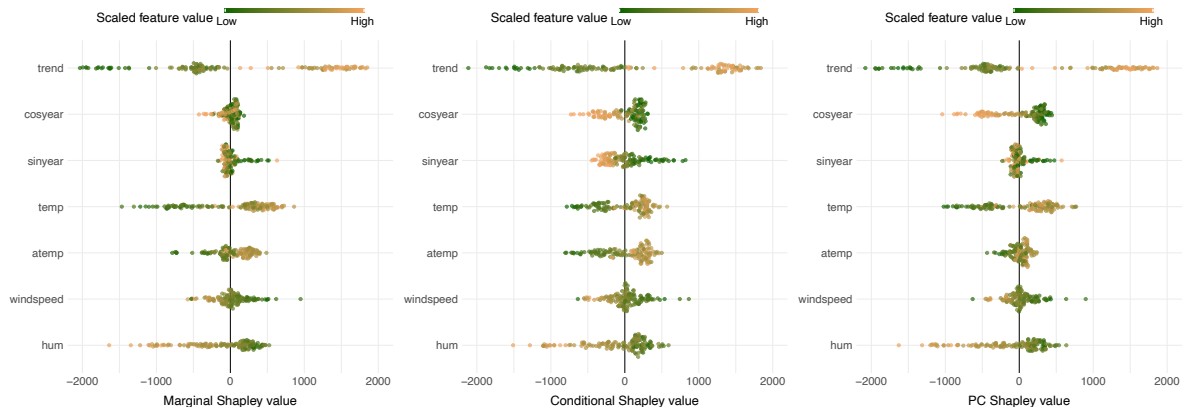

Figure 21: Sina plots for a regression task using XGBoost on the bike rental dataset. We plot marginal (left), conditional (middle) and PC Shapley values (right).

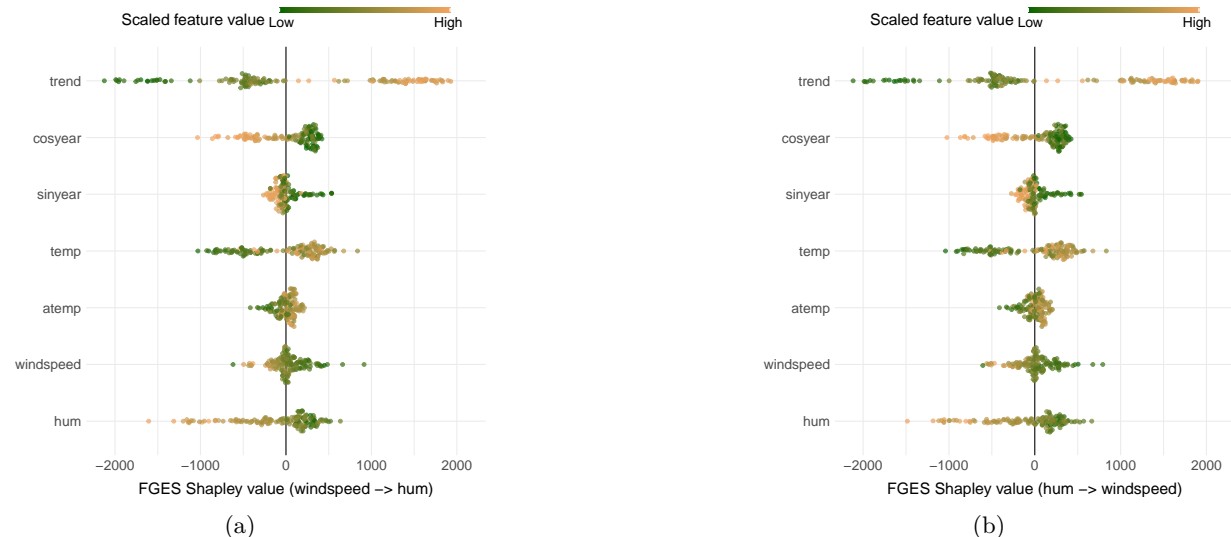

Figure 22: Sina plots for a regression task using XGBoost on the bike rental dataset. We plot FGES shapley values for the two DAGs in the CPDAG discovered by FGES (shown in Fig. 20 on the left).

we can observe slightly more spread in temperature, which is consistent with trend being an additional cause in the FGES CPDAG. In Fig. 7 we show the bar plot of the Shapley values for two days with temperature about 13 degrees Celsius in October and December. While this temperature represents an average day in October, it is a warm day for December, hence the contribution of the temperature to the prediction of bike rental is substantially more positive for December for both Causal chain and PC Shapley values. Similarly, there is a significant seasonal drop in bike rentals from October to December, which can be seen in both Causal Chain and PC Shapley values, while this does not hold true for marginal Shapley values. Finally, similarly to what is shown in the sina plot, PC Shapley values attribute a bigger impact to temperature than Causal chain Shapley values.

## A.5 Implementation and compute

To implement our method, we adapted the 'shapr' package by Aas et al. (2021) (https://github.com/NorskRegnesentral/shapr) released under the MIT license. The causal discovery methods we used were implemented in the 'pcalg' package (Kalisch et al., 2012) released under GPL-2, the 'kpcalg' package (Zhang et al., 2011) released under GPL2, and the Tetrad package (Ramsey et al., 2018) released under the GPL-2

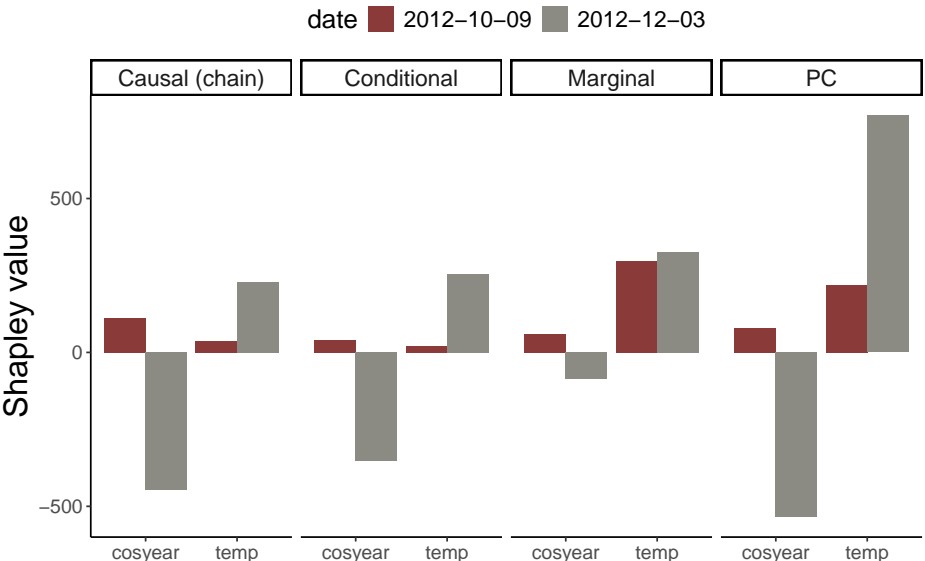

Figure 23: Bar plot of Shapley values for the features *cosyear* and *temp* of the bike rental dataset for two days in October and December with the same temperature. We show the Causal chain Shapley values from Heskes et al. (2020), along with marginal, conditional, and PC Shapley values. Both FGES Shapley values are similar to PC and have been omitted.

license. We make use of the 'igraph' package (Csardi & Nepusz, 2006), released under the GPL license, to represent graphs. To identify causal effects we make use of the 'causaleffect' package (Tikka & Karvanen, 2017a) released under GPL-2. The prediction models we trained were xgboost models (Chen & Guestrin, 2016) released under the Apache License v2.0. For estimating conditional Gaussian distributions we make use of the 'mvnfast' package (Fasiolo, 2014), released under GPL-2. For analysis of results and the creation of plots we make use the 'tidyr', 'dplyr' and 'ggplot2' packages, all part of the tidyverse (Wickham et al., 2019), released under the MIT license. We provide the code to reproduce all our experiments in the supplementary material.

### A.5.1 Hyperparameters

In this section we list each hyperparameter used in the experiments, along with an explanation, how we vary them, and their default value in bold.

When sampling random DAGs, we specify $p$ the amount of nodes excluding the dependent variable (Y), $d$ the expected number of neighbors for each node $X_i$ (representing features), and $\bar{m}$, the expected amount of parents of Y, i.e. the expected number of features directly affecting the dependent variable. We use $p = 5, \mathbf{10}, 15$, $d = 1, \mathbf{2}, 3$, and $\bar{m} = 3, \mathbf{6}, 9$ for $p = 10$.

We consider two parameterizations for each graph: linear Gaussian and non-linear. In the linear Gaussian setting, nodes are parametrized as $X_i = \sum_{j \in \mathrm{Pa}_i} w_{ij} X_j + \epsilon_i$ where $w_{ij} \sim U((-2, -0.5) \cup (0.5, 2))$ and $\epsilon_i = \mathcal{N}(0, 1)$.

For the non-linear setting we consider parameterizations of the form $X_i = w_2^i \sigma(\sum_{j \in \mathrm{Pa}_i} w_1^{ij} X_j) + \epsilon_i$ where $\sigma$ denotes the sigmoid function, $w_1^{ij} \sim U((-1.5, -0.5) \cup (0.5, 1.5))$, $w_2^i \sim U((-3, 1) \cup (1, 3))$, and $\epsilon_i$ follows either a $\mathcal{N}(0, 1)$ or a $U(-1, 1)$ distribution with equal probability. As such, for each DAG we get two different SCMs, one linear Gaussian and one non-linear. For each and each parameterization we sample $n_{\mathrm{scm}} = 40$ random SCMs.

For each setting of hyperparameters we generate $n_{\mathrm{scm}} = 40$ random causal models.

For causal discovery, we use the PC method implemented in the 'pcalg' package (Kalisch et al., 2012). We set a default significance level of $\alpha = 0.05$. We use the default settings, which uses PC-stable by Colombo & Maathuis (2014). For the linear Gaussian parameterizations we use partial correlation conditional independence tests ('gaussCI') and for the non-linear setting kernel-based conditional independence tests implemented by the 'kpcalg' package (Zhang et al., 2011), specifically the Hilbert-Schmidt Independence Criterion gamma test. For FGES we use the TETRAD toolbox (Ramsey et al., 2018) using the BIC score with default penalty discount 1. We use the following values for number of observational samples that we apply PC and FGES to $n_{cd} = 250, 500, \mathbf{1000}, 2000, 4000$.

Prediction models were trained using XGBoost (Chen & Guestrin, 2016) using $\eta = 0.1$, and 'subsample = 0.8' for a 100 rounds. We used $n_{train} = 10K$ observational samples to train each model.

For computing Shapley values we consider the hyperparameters $n_{comb}$, $n_{mc}$, $n_{obs}$. $n_{comb}$ specifies the number of combinations used by all methods, where we use the full set of combinations for $p = 5, 10$, and for $p = 15$ $n_{comb}^{true} = 8192$ for the ground truth and $n_{comb} = 4096$ for the other methods. $n_{mc}$ specifies the number of Monte Carlo samples used to estimate each expectation. We use the values $n_{mc} = 250, 500, \mathbf{1000}, 2000, 4000$, with $n_{mc}^{true} = 4000$ fixed for the ground truth. $n_{obs}$ specifies the amount of observational datapoints used to estimate marginal and conditional distributions, which we vary $n_{obs} = 250, 500, \mathbf{1000}, 2000, 4000$.

For each SCM we evaluate performance on $n_{test} = 40$ data points unseen by the prediction model.

### A.5.2   Computing infrastructure

The full experiment suite was run in parallel on a cluster of AMD Rome CPUs with 16 cores each. Total compute time for all experiments listed was approximately 88 days. Individual runs in the default setting per 40 data points varied from 30 seconds (marginal, conditional), up to 4 hours for MECs of the largest size. Runs for 15 nodes could take up to 8 hours for 40 datapoints.

