# OpenReview forum: "Improving Local Explainability By Learning Causal Graphs From Data"
_TMLR — Accepted by TMLR_

### Review · Reviewer_Su8y · 2025-12-15

**Summary Of Contributions:**

The paper focuses on overcoming the limitation of existing causal shapley value methods that leverage only known causal relations. The main contributions of the paper are as follows:
- The authors propose an efficient framework that combines causal discovery methods with the shapley value methods, to yield a set of explanations for the datapoints.
- The paper presents two variants of the framework: sampling based and importance weighting based.
- The experiments on both simulated and real-world datasets are comprehensive covering variations like varying degree in the graph, number of parents of the prediction variable and ablation studies for the hyper-parameters. The results demonstrate the effectiveness of the method in estimating the shapley value as compared to the conditional and marginal shapley value baselines.

**Key Strengths:**

- The paper is well-written and clearly lays down the algorithm and mathematics behind the proposed algorithm.
- The experiments are technically sound and comprehensive. The time analysis for the different methods is interesting.

**Key Weaknesses:**
- The experiment methodology is unclear at certain points. It is unclear what the bars in the graphs for different methods indicate. Please see below for more details.

**Audience:**

Yes

**Audience Explanation:**

The idea of combining shapley value estimation with causal discovery is very natural and helpful. The authors also present initial ideas for tackling the computational complexity of the problem as well. Therefore, it could be a good starting point for the people working on explainability in diverse real world disciplines.

**Broader Impact Concerns:**

Most concerns are already addressed.

**Claims And Evidence:**

No

**Claims Explanation:**

I have concerns about the claim that the proposed method can still output largely close feature shapley values to the ground truth/oracle values despite the incorrectly learned graph.

In synthetic cases, this claim is supported by the numbers of the proposed method. However, on the real world dataset, the PC MEC method performs quite worse and this behaviour is attributed to the incorrectly learned edge between two variables. The two observations don’t align together as if the model were robust to minor mistakes in the graph, it would be reflected in the real world case as well.

Theoretically, I would expect the model to give abnormal values to the incorrect graphs depending on the SHD rather than covering the underlying mistake with close shapley values. These abnormal values could then help in identifying incorrect graphs and even localize to what relations might be oriented wrong. A discussion from the authors on this would be helpful.

**Requested Changes:**

There are 2 adjustments which I would like to propose:
- A description of the graphs is necessary, especially where the spread is coming from. If the spread is coming from the different graphs in the MEC, then why would the oracle MEC have the spread. A clear description of what the plotted values are and what the bars are will make it more clear.
- The claim about robustness to incorrect graphs needs to be justified with regards to why it is required and what it intends to achieve. A plot with NL2E on y and SHD on x axis could be interesting to see what the tolerance of the algorithm is relative to the size of the problem. Consistency across both synthetic and real world datasets is a must.

While the second adjustment is critical to securing my recommendation for acceptance, the first one is slightly less critical but still important for clear messaging and hence acceptance.

---

> ### Author Response · Authors · 2026-02-04
>
> Thanks for your careful review and encouragement. We address your concerns in the following and in the revised version of the paper, where we have highlighted any major change in purple. Additionally, we ask for clarification regarding one of your points.
>
> ## Clarification of plots and metrics
>
> Thanks for the comments, we added these clarifications to the revised version of the paper, both in the figure captions and more in detail in the text in the Metrics and Results paragraph on pages 8 and 9.
>
> In short, the bars are representing what we now formally define as the “NL2E over the range of explanations” for a MEC Shapley value method, i.e., the minimum and maximum of the NL2E over the DAGs in the MEC returned by each method. In the plots this is averaged over the $n_{scm}=40$ causal models and then the bars represent the average minimum and the average maximum NL2E. The marker represents the average NL2E across the DAGs in the MEC, again averaged over all the causal models. We report the standard error across these causal models in Appendix A.3.2, showing that our evaluation is stable across the causal models.
>
> We also added an explanation to why the Oracle MEC Shapley values method has a spread, which is expected since there is still quite a lot of variability in the NL2E across the DAGs even in the ground truth MEC, since each DAG will yield different causal Shapley values. Moreover, this spread might not include the ground truth Causal Shapley values, because the Oracle MEC method has only access to the oracle MEC, but still uses the same $n_{obs}$ finite sample data as the other methods to actually estimate the conditional distributions.
>
> ## Clarification of the benefits of the methods despite the incorrectly learned graphs
>
> Thanks for raising this point, we agreed that our claims should be qualified.  In particular, we revised the paper to clarify that we intend that the average NL2E for the MEC Shapley values is on average lower than most baselines in most settings. This means that while learning an incorrect structure might still provide on average a useful explanation in the simulated settings, where we average over multiple causal models, in the real-world cases with a single causal model that might not be true.
>
> Moreover, we have clarified the specific structural differences between the CPDAGs in both real-world settings, which are not limited to one edge, as previously suggested by our text.
>
> Instead, we have clarified that in ADNI the CPDAG learned by PC only identifies correctly two out of five edges, while the one learned by FGES has four correct edges out of five. This is a substantial difference in graphical structure, so it also explains the difference in output.
>
> In the bike dataset, we do not have access to the ground truth causal graph, so we can only reason about the causal relations from a qualitative point of view. Moreover, many variables are deterministic relations of others, e.g. cosyear and sinyear, which is a violation of the causal faithfulness condition, and there might be latent confounding, so the methods we use might be misspecified. In this setting, we clarify that FGES uncovers 9 edges, most of them reasonable based on common-sense knowledge, while PC misses two of these edges and orients two edges differently. In this case, most of the structure seems to be similar for the two methods, especially for the most relevant features for prediction, so combining them with the MEC Shapley value approach leads to very similar results.
>
> ## New results on SHD vs NL2E
>
> To directly quantify how discovery error relates to explanation error beyond this within-MEC variability, we additionally included the requested NL2E-versus-SHD analysis in the revised version of the paper in Figure 17 in the Appendix for linear Gaussian causal models with $p=10$, $d=2$, so on average 20 edges. The labels on the points indicate how many CPDAGs fit the specific SHD level.
>
> As now discussed in Appendix A3.10, these results show that as SHD grows, NL2E tends to increase on average, albeit with substantial spread at each SHD level. Interestingly, the NL2E spread across the DAGs in each MEC does not seem to increase predictably with higher SHD. One explanation is that the spread is due to the size of the MEC, which is known to be asymptotically constant and empirically on average containing 4 DAGs [Gillispie et al 2001].
>
> ## Abnormal values
>
> In real-world settings, we do not know what the “normal” or “abnormal” causal Shapley values look like, so we cannot use this signal to somehow falsify incorrect models, let alone specific causal relations. In particular, we cannot really use the spread between the explanations to falsify the models, since as shown in our experiments in Figure 17,  there is spread across the NL2E values even in the ground truth MEC due to the differences across all the DAGs in the MEC (so with SHD=0), and this spread does not seem to increase predictably with more incorrect CPDAGs.

---

> > ### Comment · Reviewer_Su8y · 2026-02-07
> > **Thanking the authors for the revision**
> >
> > Thank you for the revised submission. The revisions clarify most of my concerns and add the required missing details.
> >
> > Figure 17 is particularly interesting, showing that the within-MEC spread remains nearly the same irrespective of the SHD.

---

### Review · Reviewer_z4iG · 2025-12-22

**Summary Of Contributions:**

This manuscript looks at the potential of using causal discovery to improve explainability analysis through shapley values.  In particular, they focus on the case where the causal graph is not known a priori and needs to be estimated from data, and how that can be used to modify the structure put into estimating the shapley values.

Overall, the manuscript is clearly written with a clear idea.  The algorithm is straightforward to implement although computationally challenging in larger dimensions.  There is a clear advantage over the chosen baselines in non-linear scenarios, although advantages in linear scenarios are less clear.

The biggest weakness to me is that it is not clear when this would provide substantially different or better results than correcting for the relationships between the variables; e.g., do we need the causal viewpoint, or do we just need to estimate relationships between variables better?  The chosen baselines are fairly minimal, considering standard (independent) shapley analysis and correcting for a linear structure--or so I assume, as the algorithmic details on the baseline are not particularly precise.  I would be more convinced by the approach is there was clear theoretical analysis about when the results will diverge (e.g., are they essentially the same in the linear gaussian case?) and if there was greater attempts to capture non-linear relationships between variables in a baseline model without causality.  This, to me, is addressable.

The other weakness is the challenges of scalability; the primary shown results are based on relatively small and not particularly connected graphs with a large number of measurements.  While that certainly covers a subset of problems, the majority of ongoing scientific challenges that I interact with have fewer measurements and more variables, making it unclear how broadly applicable this method is.  I would encourage the authors to consider scaling to a greater extent, which is challenging with the causal discovery approaches.

**Audience:**

Yes

**Audience Explanation:**

Shapley analysis is a commonly used method, and improvements in its efficacy would be appreciated.

**Broader Impact Concerns:**

None.

**Claims And Evidence:**

No

**Claims Explanation:**

The first gap in the evidence that I see is that that there is not a clear enough evidence about when this approach would outperform a conditional method in a linear-gaussian case, as the results are largely overlapping.  I would appreciate this being examined from both a theoretical and practical perspective.  In particular, is it just that the two-step method (causal discovery + interventional effects) has a more robust estimation of the correlation structure?

The second gap is that there is not a clear enough description of the baseline conditional approach and how it is implemented.  There are several different versions of this approach, so details are necessary.  It is somewhat implied that it is the linear formulation, which would make sense as for why it does not capture non-linear relationships.  If so, then it should be considered what happens when you capture non-linear conditional relationships.

Additionally, it would be helpful to visualize the Sina plots in the main draft as to show how the estimates change.  For example, Figure 16 is helpful for understanding the differences between the methods; however, there the primary difference is that it captures indirect effects.  Given that the marginal method cannot capture indirect effects, it would be informative to see what the conditional results show differently there.  That would help to give a sense of the practical importance of the method.

**Requested Changes:**

I would like the authors to:
(a) evaluate mathematically when the proposed approach differs from a conditional estimation.
(b) provide more details on the baseline algorithms (how were the conditional relationships estimated)?
(c) evaluate how different conditional estimation procedures impacts results
(d) visualize shapley values to compare against all methods
(e) ensure all details are included on data sets (e.g., sample numbers, etc., on the real-world data)

I think these are all necessary.

---

> ### Author Response · Authors · 2026-02-04
> **Response 1/2**
>
> Thanks for your detailed review and constructive comments. We answer your concerns one by one in the following and in the revised version of the manuscript, where we have highlighted any major change in purple.
>
> ## a) Mathematical analysis showing how it is different than conditional Shapley
>
> We have added a new Appendix A2 in the revised version of the paper, in which we provide:
>
> 1. A mathematical analysis with two Lemma and a Corollary characterizing the special case in which conditional and causal Shapley values coincide. The final Lemma shows that they will be identical and all their value functions will be identical if the features are independent, since an interventional and conditional distribution are in general identical if the intervened variables are non-descendants of the others, which would require all features to be non-descendants of each other when we iterate over all subsets $S \subseteq [p]$.
>
> 2. A worked out example of what happens in a linear gaussian case where the causal graph is a **v-structure**, a basic block of causal discovery. The Markov Equivalence Class of a v-structure contains only one DAG, the v-structure itself. We show that even in this simple case, the conditional and causal Shapley values differ substantially. In particular, the conditional Shapley values also condition on the descendants of a feature, introducing bias similar to selection bias in the explanations. Moreover, by ignoring the causal order, they also assign non-zero Shapley values to two features that have no effect (direct or indirect) on the prediction.
> In this example we used the ground truth values of the parameters of the linear model, so this shows that the difference is not only about better estimation, but actually about the causal modeling.
>
> Since in general causal discovery is able to identify at least some of the causal edges (e.g. the v-structures in larger graphs, as well as other orientations that might propagate from them), this minimal example showcases how our approach can use the inductive bias of having a causal graph to provide a better output than conditional Shapley values. Moreover, since our approach outputs a list of causal Shapley values, including in the oracle case the ground truth one, this is still a more informative explanation than the single output of the conditional Shapley values.
>
> In finite sample settings, as in our experiments in the linear gaussian case, this improvement can be potentially limited by the difficulty of learning the correct causal relations with the causal discovery methods we considered. In Figure 2 and especially Figure 3a, the Oracle MEC Shapley values seem to consistently provide a range of explanations that is on average better than the conditional Shapley values. This is particularly visible in the non-linear setting, both in the figures in the main paper, in which we use a Gaussian approximation for all methods, but also in the new results in Appendix A.3.3, where we also consider more expressive non-linear estimators for the baselines.
>
> ## b) More details on baselines
>
> We have updated Section 5.1 to explicitly state that our conditional baseline employs the Gaussian approximation from Aas et al. (2021). We clarify that the conditional distributions $P(X_{\bar{S}}|X_S = x_s)$ are derived analytically from the observational sample mean and covariance and that we use the same conditional estimator for conditional and MEC Shapley values to allow for a fair comparison.
>
> We selected this specific formulation as our primary baseline to ensure comparability with prior work in the causal Shapley literature (e.g., Heskes et al., 2020; Aas et al., 2021). This allows us to isolate the specific contribution of our causal discovery framework, separate from the choice of conditional estimator. As you correctly inferred, this formulation assumes linear relationships, but we provide additional results with non-linear estimators that we explain in the next response.
>
> ## c) Different conditional estimation for baselines, especially non-linear
>
> We appreciate the suggestion to evaluate conditional Shapley with a more expressive conditional estimator in the non-linear setting. We have implemented this using the `shapr` framework’s copula-based conditional estimator (continuous features) and the conditional inference tree estimator, which potentially better captures non-linear dependence than a Gaussian conditional model. We add results for the baselines in Appendix A.3.3. The copula baseline yields only a marginal improvement over the Gaussian conditional baseline, and our proposed method, despite still using the Gaussian conditional estimation, remains on average closer to the true causal Shapley values. This suggests that the observed gains are not primarily driven by conditional model misspecification, but by the causal structure used by our method.

---

> > ### Author Response · Authors · 2026-02-04
> > **Response 2/2**
> >
> > ## d) Sina plots in the main draft for all methods.
> >
> > We have shortened a few words in the revised version of the manuscript and slightly reduce the figures sizes, so we could add the sina plot of the Alzheimer’s dataset for the conditional Shapley values, the ground truth causal Shapley values and FGES Shapley values in the main paper in Figure 6. The sina plots for all methods are in Figure 19 in the Appendix. We also added to the Appendix the sina plots for the two DAGs discovered by FGES on the bike rental dataset (Figure 22).
> >
> > ## e) More dataset details
> >
> > We added explicit counts of the number of data points and number of variables to sections A.4.1 (Alzheimers) and A.4.2 (bike rental) of the Appendix. Pre-processing and train/test splits were already detailed.
> >
> > ## Scalability
> >
> > We agree with the limitations of our methods in terms of scalability and added an explanation in the revised manuscript. In short, our method is limited by the large computational cost of causal discovery, which is notoriously an NP-hard problem in general, but even more so Shapley estimation (which is exponential without coalition subsampling). Our methods improve on the scalability by reusing computations across coalitions and graphs in the MEC, but scaling them beyond a few tens of variables might require considering a completely novel approach.

---

> > > ### Comment · Reviewer_z4iG · 2026-02-16
> > >
> > > Thank you for your clarifications.  I think the revision has improved the manuscript and the clarity of the message.

---

### Review · Reviewer_EMdU · 2026-01-21

**Summary Of Contributions:**

First of all, as a reviewer within the realm of causal AI, it is important to thank the authors for their cool project. Any time invested into the philosophy, science and engineering to make systems explainable in a causally consistent manner that can actually be deployed widely is awesome and much appreciated.

The authors present in their manuscript a complete pipeline that goes from data (purely associational) to learning different possible causal structures to then give causal explanations based on those all while respecting the fact that this requires re-use of computation to efficiently get said explanations (since we observe [super]exponential behavior for both DAG possibilities and MEC size). A derivation for the estimation methods is provided. Extensive experimentation is provided covering different data sets, settings and ablating over important parameters. An appendix with additional discussion is provided. Further code for reproduction is provided (not evaluated).

**Additional Comments:**

Ideally the authors can extend their experimentation. More data sets, more CD methods.

**Audience:**

Yes

**Audience Explanation:**

My judgment on this is that likely anyone interested in XAI or causal AI will find this paper interesting and might even consider using it in their paper.

**Broader Impact Concerns:**

Explanations and sound reasoning are crucial in our current age of AI. IMHO this type of research will benefit all of us greatly down the road. Other than that the paper does not raise any concerns. My hope lies in the continuation of such research.

**Claims And Evidence:**

Yes

**Claims Explanation:**

A key challenge for causal Shapely values, which poses the key method the authors build upon, is the requirement of the (true) causal graph. This is solved by simply attaching a causal discovery process a priori to the pipeline that learns from the same data the predictor-to-be-explained was trained on. This naturally comes with challenges of both design choices regarding performance (accuracy, time and such). The authors focus on a robust pipeline that re-uses computation to be efficient. The authors present two estimation methods: 1) using sampling where interventional data is generated using a candidate graph given the TF-formula, and 2) using IW where ID is used with the graph to get an estimand applicable to the available data. The key technical contribution is the grouping of graphs into sets that share interventional distributions and said estimation methods for the interventional expectation in Eq.4. While said contribution might be seen as only slightly incremental, the execution both technically and presentation-wise are solid. Furthermore, a proper study is presented that gives a reproducible account of our topic at hand. Reproducibility, precise language, interesting findings, important problem, mechanistic explanations - in terms of scientific endeavour everything checks out.

**Requested Changes:**

IMHO the main changes to this particular submission are of polishing nature. One could go back on any segment of the paper and make sure that they are to be understood by as many readers as possible for instance being self-contained e.g. in Alg.1 when reading something like $p$ one might have forgotten by that point that it was the symbol to refer to the total number of features or the "final" feature of our data. Or polish the key figure such that $S_i$ becomes more clear when trying to understand (particularly for the first read) the grouping of the graphs. Things of that nature essentially, which there are many of that one can come up with to make the paper "perfect".

---

> ### Author Response · Authors · 2026-02-04
>
> Thanks for your kind and encouraging words, and for acknowledging the value of our work. We have tried to address one of the concerns that you mentioned about readability, while we did not completely get the comment about Fig. 1, but we are happy to discuss it further.
>
> Additionally, we have revised the manuscript based on the other reviewers’ concerns, improving its readability and completeness. We have highlighted any major change in purple. In particular, we added some new results that we hope further clarifies our contributions also from your point of view:
> - New mathematical analysis in Appendix A2 with a worked out example of how (MEC) causal Shapley values provide a better explanation than conditional Shapley values for a v-structure;
>
> - New results in Appendix A.3.3 for two types of conditional estimators for the non-linear case: Gaussian copula and Conditional Inference Trees, showing similar results to the main paper;
>
> - New results in Appendix A3.10 showing that as expected, the NL2E increases with higher SHD errors.
>
> We are happy to discuss further if you have any questions or comments.

---

> > ### Comment · Reviewer_EMdU · 2026-02-06
> > **Thanking the Authors for the Revision**
> >
> > Thank you.

---

### Author Response · Authors · 2026-01-21
**Thank you! Detailed responses after the ICML submission deadline**

Dear reviewers and AE,

thanks a lot for your reviews and your encouraging words.
Addressing your concerns will surely improve our paper.

Since we are quite busy with the ICML submission deadline and we expect you might be as well, we will be posting a detailed response for each of your reviews and an updated document only after the ICML deadline on Jan 28 at 23:59 AOE.

We just wanted to let you know about this in advance, so you are not wondering what is happening on our end.
We hope this also fits well with your schedule.

Cheers,
The authors

---

### Decision · Action_Editor_ydxv · 2026-02-16

**Recommendation:** Accept as is

**Audience:**

Yes

**Audience Explanation:**

The work is relevant to TMLR readers in explainability and causal ML because it tackles the common setting where the causal graph is unknown and integrates causal discovery with causal Shapley-style explanations in a principled way.

**Claims And Evidence:**

Yes

**Claims Explanation:**

The revised paper supports its main claims with clear methodology and extensive experiments on synthetic and real-world datasets. The authors also addressed prior concerns by (i) adding theoretical analysis clarifying when causal vs conditional Shapley differ, (ii) clarifying baseline implementations and adding stronger non-linear conditional baselines, and (iii) improving metric/plot explanations and analyzing robustness vs causal-discovery error. Remaining scalability/assumption limits are acknowledged and do not undermine the claims as stated.